# Activated $\alpha_{IIb}\beta_3$ on platelets mediates flow-dependent NETosis via SLC44A2

**Adela Constantinescu-Bercu[1,2], Luigi Grassi[3,4,5], Mattia Frontini[3,4,6], Isabelle I Salles-Crawley[1†]\*, Kevin Woollard[2†]\*, James TB Crawley[1†]\***

[1]Centre for Haematology, Department of Immunology and Inflammation, Imperial College London, London, United Kingdom; [2]Centre for Inflammatory Disease, Department of Immunology and Inflammation, Imperial College London, London, United Kingdom; [3]Department of Haematology, University of Cambridge, Cambridge Biomedical Campus, Cambridge, United Kingdom; [4]National Health Service Blood and Transplant, Cambridge Biomedical Campus, Cambridge, United Kingdom; [5]National Institute for Health Research BioResource, Rare Diseases, Cambridge University Hospitals, Cambridge, United Kingdom; [6]British Heart Foundation Centre of Excellence, Cambridge Biomedical Campus, Cambridge, United Kingdom

**\*For correspondence:**
i.salles@imperial.ac.uk (IIS-C);
k.woollard@imperial.ac.uk (KW);
j.crawley@imperial.ac.uk (JTBC)

[†]These authors contributed equally to this work

**Competing interests:** The authors declare that no competing interests exist.

**Abstract** Platelet-neutrophil interactions are important for innate immunity, but also contribute to the pathogenesis of deep vein thrombosis, myocardial infarction and stroke. Here we report that, under flow, von Willebrand factor/glycoprotein Ibα-dependent platelet 'priming' induces integrin $\alpha_{IIb}\beta_3$ activation that, in turn, mediates neutrophil and T-cell binding. Binding of platelet $\alpha_{IIb}\beta_3$ to SLC44A2 on neutrophils leads to mechanosensitive-dependent production of highly prothrombotic neutrophil extracellular traps. A polymorphism in *SLC44A2* (rs2288904-A) present in 22% of the population causes an R154Q substitution in an extracellular loop of SLC44A2 that is protective against venous thrombosis results in severely impaired binding to both activated $\alpha_{IIb}\beta_3$ and VWF-primed platelets. This was confirmed using neutrophils homozygous for the *SLC44A2* R154Q polymorphism. Taken together, these data reveal a previously unreported mode of platelet-neutrophil crosstalk, mechanosensitive NET production, and provide mechanistic insight into the protective effect of the *SLC44A2* rs2288904-A polymorphism in venous thrombosis.

## Introduction

To fulfil their hemostatic function, platelets must be recruited to sites of vessel damage. This process is highly dependent upon von Willebrand factor (VWF). Upon vessel injury, exposed subendothelial collagen binds plasma VWF via its A3 domain (*Cruz et al., 1995*). Elevated shear, or turbulent/disturbed flow, then unravels tethered VWF and exposes its A1 domain, facilitating specific capture of platelets via glycoprotein (GP)Ibα.

As well as capturing platelets under flow, the A1-GPIbα interaction also induces shear-dependent signaling events (*Bryckaert et al., 2015*). For this, GPIbα first binds the A1 domain of immobilized VWF (*Zhang et al., 2015*). Rheological forces then cause unfolding of the GPIbα mechanosensitive domain that translates the mechanical stimulus into a signal within the platelet (*Ju et al., 2016*; *Zhang et al., 2015*). This leads to release of intracellular $Ca^{2+}$ stores and activation of the platelet integrin, $\alpha_{IIb}\beta_3$ (*Gardiner et al., 2010*).

VWF-mediated signaling transduces a mild signal. Consequently, these signaling events are often considered redundant within hemostasis as platelets respond more dramatically to other agonists present at sites of vessel injury (e.g. collagen, thrombin, ADP, thromboxane A2) (*Jackson et al.,*

**eLife digest** Platelets in our blood form clots over sites of injury to stop us from bleeding. Blood clots can also occur in places where they are not needed, such as deep veins in our legs or other regions of the body. Developing such clots – also known as deep vein thrombosis (or DVT for short) – is one of the most common cardiovascular diseases and a major cause of death. Although certain inherited factors have been linked to DVT, the underlying mechanisms of the disease remain poorly understood.

In addition to platelets, the pathological (or dangerous) clots that cause DVT also contain immune cells called neutrophils which fight off bacterial infections. Platelets are recruited to the wall of the vein by a protein called "von Willebrand Factor" (or VWF for short). However, it remained unclear how these recruited platelets interact with neutrophils and whether this promotes the onset of DVT.

To answer this question, Constantinescu-Bercu et al. used a device that mimics the flow of blood to study how human platelets change when they are exposed to VWF. This revealed that VWF 'primes' the platelets to interact with neutrophils via a protein called integrin $\alpha_{IIb}\beta_3$. Further experiments showed that integrin $\alpha_{IIb}\beta_3$ binds to a protein on the surface of neutrophils called SLC44A2. Once the neutrophils interacted with the 'primed' platelets, they started making traps which increased the size of the blood clot by capturing other blood cells and proteins.

Finally, Constantinescu-Bercu et al. studied a genetic variant of the SLC44A2 protein which is found in 22% of people and is associated with a lower risk of developing DVT. This genetic mutation caused SLC44A2 to interact with 'primed' platelets more weakly, which may explain why people with this genetic variant are protected from getting DVT.

These findings suggest that blocking the interaction between 'primed' platelets and neutrophils could reduce the risk of DVT. Although current treatments for DVT can prevent patients from forming dangerous blood clots, they can also cause severe bleeding. Since neutrophils are not crucial for normal blood clots to form at the site of injury, drugs targeting SLC44A2 could inhibit inappropriate clotting without causing excess bleeding.

*2003*; *Senis et al., 2014*). Full platelet activation involves release of α- and δ-granules, presentation of new cell surface proteins, activation of cell surface integrins and alterations in the membrane phospholipid composition. The extent of platelet activation is dependent upon both the concentration, and identity, of the agonist(s) to which the platelets are exposed, which is dictated by the location of the platelets relative to the damaged vessel. For example, platelets in the core of a hemostatic plug/thrombus are exposed to higher concentrations of agonists and are more highly activated (i.e. P-selectin-positive procoagulant platelets) than those in the surrounding shell (P-selectin-negative) (*de Witt et al., 2014*; *Shen et al., 2017*; *Stalker et al., 2013*; *Welsh et al., 2014*). Thus, platelets exhibit a 'tunable' activation response determined by agonist availability.

Aside from hemostasis, platelets also have important roles as immune cells by aiding in targeting of bacteria by leukocytes (*Gaertner et al., 2017*; *Kolaczkowska et al., 2015*; *Sreeramkumar et al., 2014*; *Wong et al., 2013*). Platelet-leukocyte interactions also influence the development of inflammatory cardiovascular conditions. In deep vein thrombosis (DVT), VWF-dependent platelet recruitment, platelet-neutrophil interactions and the production of highly thrombotic neutrophil extracellular traps (NETs) all contribute to the development of a pathological thrombus (*Brill et al., 2011*; *Brill et al., 2012*; *Fuchs et al., 2012a*; *Schulz et al., 2013*; *von Brühl et al., 2012*). Although the precise sequence of events still remains unclear, it appears that during the early stages of DVT, VWF-bound platelets acquire the ability to interact with leukocytes (*von Brühl et al., 2012*). Exactly how this is mediated given the lack of vessel damage is unclear. It also remains to be determined precisely how platelet-tethered neutrophils undergo NETosis in DVT in the absence of an infectious agent.

Known direct platelet-leukocyte interactions involve either P-selectin or CD40L on the surface of platelets binding to P-selectin glycoligand-1 (PSGL-1) and CD40, respectively, on leukocytes (*Lievens et al., 2010*; *Mayadas et al., 1993*; *Palabrica et al., 1992*). As platelets must be potently activated to facilitate P-selectin/CD40L exposure, such interactions unlikely mediate the early

platelet-leukocyte interactions that occur in the murine DVT model. Consistent with this, lack of platelet P-selectin has no effect upon either leukocyte recruitment or thrombus formation in murine DVT (*von Brühl et al., 2012*). Leukocytes can also indirectly interact with platelets through Mac-1 (integrin $\alpha_M\beta_2$), which can associate with activated $\alpha_{IIb}\beta_3$ via fibrinogen (*Weber and Springer, 1997*), or directly via GPIb$\alpha$ (*Simon et al., 2000*). Interactions are also possible through lymphocyte function-associated antigen 1 (LFA-1/integrin $\alpha_L\beta_2$) that can bind intercellular adhesion molecule 2 (ICAM-2) on platelets (*Damle et al., 1992*; *Diacovo et al., 1994*). In both instances though, leukocyte activation is necessary to activate Mac-1 or LFA-1 integrins before interactions can occur.

Although it is often assumed that only activated platelets bind leukocytes, recent studies have revealed that platelets captured under flow by VWF released from activated endothelial cells can recruit leukocytes (*Doddapattar et al., 2018*; *Zheng et al., 2015*). If VWF-GPIb$\alpha$-dependent signaling is capable of promoting leukocyte binding, this may be highly relevant to the non-hemostatic platelet functions (particularly when other agonists are not available/abundant), but may also provide major mechanistic insights into the early recruitment of leukocytes during the initiation of DVT.

Genome wide association studies (GWAS) on venous thromboembolism (VTE) have identified a panel of genes (*ABO, F2, F5, F11, FGG, PROCR*) with well-described influences upon coagulation and thrombotic risk, as well as those with well-established causative links (e.g. *PROS, PROC, SERPINC1*) (*Germain et al., 2015*; *Germain et al., 2011*; *Rosendaal and Reitsma, 2009*). This is consistent with the efficacy of therapeutic targeting of coagulation to protect against DVT with anticoagulants (*Chan et al., 2016*). However, although the use of anticoagulants is effective, dosing and efficacy are limited by the increase in the risk of bleeding in treated individuals (*Chan et al., 2016*; *Schulman et al., 2014*; *Schulman et al., 2009*; *Schulman et al., 2013*; *Wolberg et al., 2015*). Therefore, alternative targets that inhibit DVT disease processes, but that do not modify bleeding risk may provide new adjunctive therapies to further protect against the development or recurrence of DVT. GWAS studies have also identified additional risk *loci* for VTE, but with no known role in coagulation (*Apipongrat et al., 2019*; *Germain et al., 2015*; *Hinds et al., 2016*). This provides encouragement that alternative therapeutic targets may exist with the potential to modify the disease process without affecting bleeding risk. These *loci* include *SLC44A2* and *TSPAN15* genes (*Apipongrat et al., 2019*; *Germain et al., 2015*; *Hinds et al., 2016*). Despite the identification of these *loci*, the function of these cell surface proteins with respect to their involvement in the pathogenesis of venous thrombosis remains unclear.

Using microfluidic flow channels to enable analysis of the phenotypic effects of VWF-GPIb$\alpha$ signaling under flow, we confirm the rapid activation of the platelet integrin, $\alpha_{IIb}\beta_3$. Activated $\alpha_{IIb}\beta_3$ is capable of binding directly to neutrophils via a direct interaction with SLC44A2. Under flow, this interaction transduces a signal into neutrophils capable of driving NETosis. A single-nucleotide polymorphism (SNP; rs2288904-A) in *SLC44A2* (minor allele frequency 0.22) that is protective against VTE (*Germain et al., 2015*) encodes a R154Q substitution in the first extracellular loop of the receptor that markedly reduces neutrophil-platelet binding via activated $\alpha_{IIb}\beta_3$. These results provide a functional explanation for the protective effects of the rs2288904-A SNP and highlight the potential of SLC44A2 as an adjunctive therapeutic target in DVT (*Constantinescu-Bercu et al., 2020*).

## Results

To explore the influence of platelet binding to VWF under flow upon platelet function, full length (FL-) human VWF was adsorbed directly onto microfluidic microchannel surfaces, or the isolated recombinant VWF A1 domain, or an A1 domain variant (Y1271C/C1272R, termed A1*) that exhibits a 10-fold higher affinity for GPIb$\alpha$ (*Blenner et al., 2014*), were captured via their 6xHis tag. Fresh blood anticoagulated with D-phenylalanyl-prolyl-arginyl chloromethyl ketone (PPACK) and labeled with DiOC$_6$, was perfused through channels at 1000 s$^{-1}$ for 3.5 min. On FL-VWF, A1 or A1*, a similar time-dependent increase in platelet recruitment/surface coverage was observed (*Figure 1a* and *Figure 1—figure supplements 1–2*).

Platelets rolled prior to attaching more firmly on all VWF channels. However, median initial platelet rolling velocity on VWF A1 was 1.76 µms$^{-1}$, whereas on A1* this was significantly slower (median 0.23µms$^{-1}$) (*Figure 1b & c*), reflective of its 10-fold higher affinity for GPIb$\alpha$ (*Video 1*).

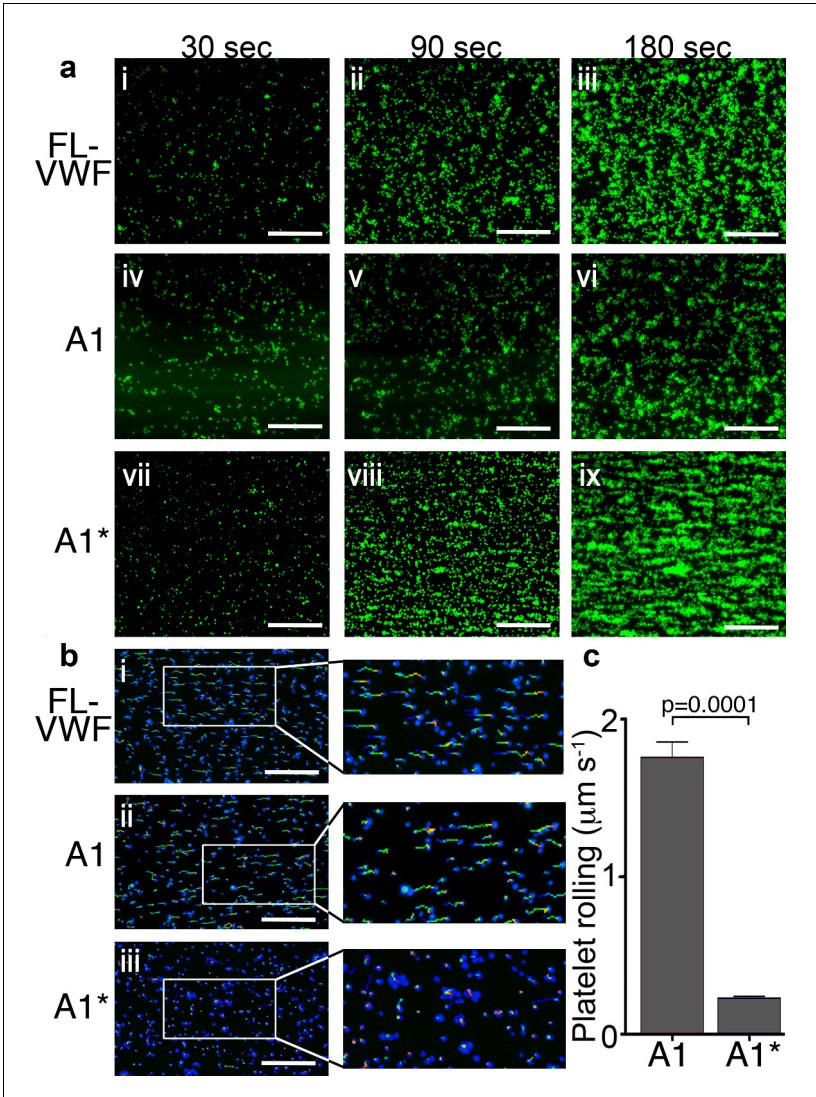

**Figure 1.** Platelet rolling and attachment to VWF under flow. (a) Vena8 microchannels were coated with either full-length VWF (FL-VWF; i-iii), VWF A1 (iv-vi) or A1* (vii-ix). Whole blood labeled with DiOC$_6$ was perfused at 1000 s$^{-1}$. Representative images (n = 3) of platelets (green) after 30, 90 and 180 s are shown. Scale; 50 μm (see also *Video 1*). (b) Experiments performed as in a), bound platelets (blue) were tracked (depicted by multi-colored lines) representing distance travelled in the first 30 s of flow. Scale bar; 50 μm. (c) Platelet rolling velocity on channels coated with A1 and A1*. Data plotted are median ±95% CI. n = 3562 platelets from 3 different experiments (A1) and n = 4047 platelets from 3 different experiments (A1*). Data were analyzed using the Mann-Whitney test. The online version of this article includes the following source data and figure supplement(s) for figure 1:

**Source data 1.** Platelet rolling source data.
**Figure supplement 1.** Analysis of purified recombinant VWF A1 and VWF A1*.
**Figure supplement 1—source data 1.** Analysis of purified recombinant VWF A1 and VWF A1* source data.
**Figure supplement 2.** Platelet coverage on VWF A1, A1* and VWF surfaces.
**Figure supplement 2—source data 1.** Platelet coverage source data.

## Platelet binding to VWF under flow induces intraplatelet signaling and activation of α$_{IIb}$β$_3$

Platelets bound to either FL-VWF, A1 or A1* formed small aggregates after about 2 min (*Figure 2ai*) due to activation of the platelet integrin, α$_{IIb}$β$_3$, and its binding to plasma fibrinogen. Consistent with this, when plasma-free blood (i.e. RBCs, leukocytes and platelets resuspended in plasma-free

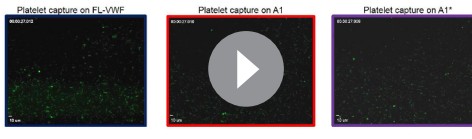

**Video 1.** Platelet capture on FL-VWF, VWF A1 or VWF A1*. Vena8 microchannels were coated with either full-length VWF (FL-VWF), VWF A1 or A1*. Whole blood labelled with $DiOC_6$ was perfused at 1000 s$^{-1}$ for 3 min. Note the reduced rate of platelet rolling on microchannels coated with A1* when compared to FL VWF or A1.

https://elifesciences.org/articles/53353#video1

buffer) was used, platelets remained as a uniform monolayer, and did not form microaggregates (*Figure 2aii*). Similarly, when activated $\alpha_{IIb}\beta_3$ was blocked in whole blood with eptifibatide or GR144053, aggregation was also inhibited (*Figure 2aiii* and iv). Irrespective of the surface (VWF, A1 or A1*), platelet aggregation was markedly reduced if plasma-free blood was used, or if $\alpha_{IIb}\beta_3$ was blocked (*Figure 2b–e*). These results demonstrate that the A1-GPIbα interaction leads to activation of $\alpha_{IIb}\beta_3$, which is consistent with previous reports (*Goto et al., 1995*; *Kasirer-Friede et al., 2004*). In support of this, fluorescent fibrinogen bound to platelets tethered via FL-VWF, but not to platelets captured to channel surfaces using an anti-PECAM-1 antibody (*Figure 2—figure supplement 1a*).

To investigate the effect of A1-GPIbα-dependent signaling, platelets were preloaded with the Ca$^{2+}$-sensitive fluorophore, Fluo-4 AM. Platelets bound to A1* under flow exhibited repeated

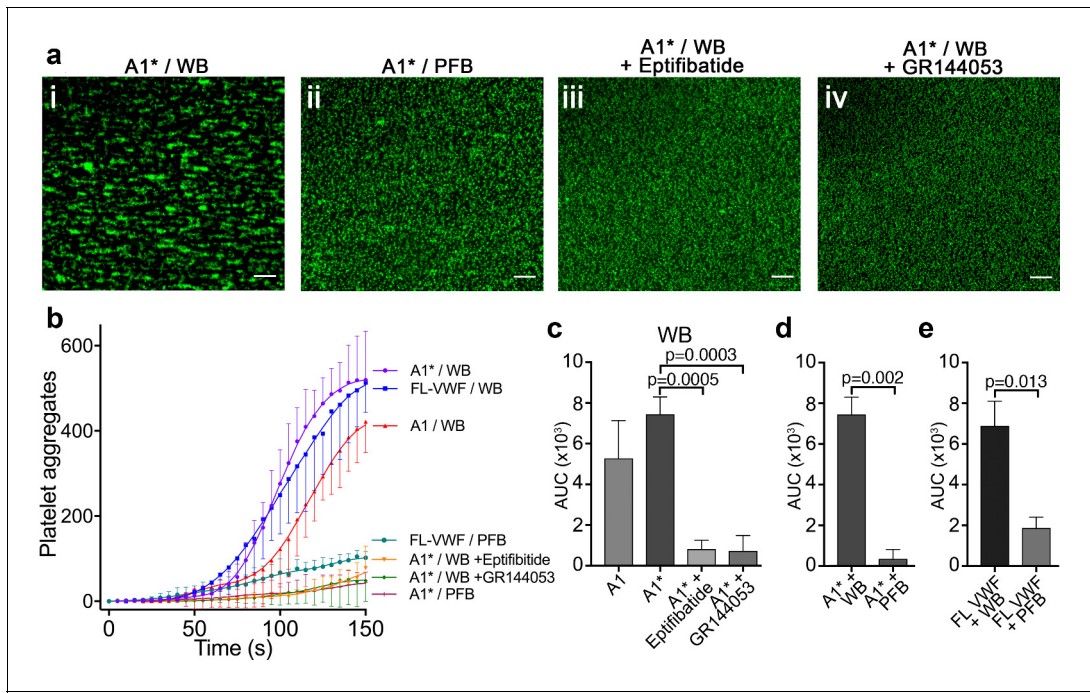

**Figure 2.** Platelet binding to VWF under flow induces $\alpha_{IIb}\beta_3$-dependent aggregation. (**a**) Vena8 microchannels were coated with A1* via its 6xHis tag. (**i**) Whole blood (WB) or ii) plasma-free blood (PFB), iii) WB containing eptifibatide or iv) WB containing GR144053 were perfused through channels at 1000 s$^{-1}$. Representative images acquired after 3 min. Scale bar; 50 μm. (**b**) Graph measuring platelet aggregation over time in WB perfused through channels coated with A1 (red, n = 3), A1* (purple, n = 4) and FL-VWF (blue, n = 3), WB pre-incubated with eptifibatide (orange, n = 3) or GR144053 (green, n = 4) over channels coated with A1* and PFB over channels coated with A1* (magenta, n = 3) or FL-VWF (teal, n = 3). Data plotted are mean ± SD. (**c–e**) Bar charts comparing area under the curve (AUC) of the data presented in b). (**c**) WB perfused over A1 or A1* with or without eptifibatide or GR144053. (**d**) WB or PFB perfused over A1*. (**e**) WB or PFB perfused over FL VWF. Data presented are mean ± SD, n = 3 or 4 as indicated in b). Data were analyzed using the Mann-Whitney test.

The online version of this article includes the following source data and figure supplement(s) for figure 2:

**Source data 1.** Platelet aggregation source data.
**Figure supplement 1.** Platelets binding to VWF under flow are 'primed' leading to activation of $\alpha_{IIb}\beta_3$, but minimal presentation of P-selectin.

transient increases in fluorescence, corresponding to Ca$^{2+}$ release from platelet intracellular stores in response to A1-GPIbα binding under flow (*Video 2*; *Kasirer-Friede et al., 2004*; *Mu et al., 2010*). Despite intracellular Ca$^{2+}$ release, this did not lead to appreciable P-selectin exposure (i.e. α-granule release) (*Figure 2—figure supplement 1b*). Intraplatelet Ca$^{2+}$ release was not detected when platelets were captured under flow using an anti-PECAM1 antibody. We therefore propose that flow-dependent VWF-GPIbα signaling 'primes', rather than activates, platelets. This 'priming' is characterized by activation of α$_{IIb}$β$_3$, but minimal α-granule release, and represents part of the tunable response of platelets.

## Platelets 'primed' by VWF interact with leukocytes

To explore the influence of platelet 'priming' upon their ability to interact with leukocytes, platelets were captured and 'primed' on VWF for 3 min at 1000 s$^{-1}$. Thereafter, leukocytes in whole blood (also labeled with DiOC$_6$) that were perfused at 50 s$^{-1}$, rolled on the platelet-covered surface (*Video 3*). Leukocytes did not interact with platelets captured via an anti-PECAM-1 antibody (*Figure 3a*), demonstrating the dependency on prior A1-GPIbα-mediated platelet 'priming'.

As VWF-'primed' platelets present activated α$_{IIb}$β$_3$, we hypothesized that 'outside-in' integrin signaling might be important for platelet-leukocyte interactions to occur (*Durrant et al., 2017*). Contrary to this, we observed a significant (~2 fold) increase in the number of leukocytes interacting with the VWF-bound platelets in plasma-free conditions (*Figure 3b*). Moreover, addition of purified fibrinogen to plasma-free blood to 50% normal plasma concentration significantly reduced platelet-leukocyte interactions (*Figure 3b*) suggesting that leukocytes and fibrinogen compete for binding 'primed' platelets. Blocking α$_{IIb}$β$_3$ (*Figure 3a–b* & *Figure 3—figure supplement 1a*) also significantly decreased platelet-leukocyte interactions irrespective of whether platelets were captured on FL-VWF or A1*, or whether experiments were performed in whole blood or plasma-free blood (*Figure 3a–c*).

To explore the role of activated α$_{IIb}$β$_3$ in binding leukocytes, platelets were captured onto anti-PECAM-1 coated channels and an anti-β$_3$ antibody (ligand-induced binding site – LIBS) that induces activation of α$_{IIb}$β$_3$ applied (*Du et al., 1993*). Antibody-mediated activation of α$_{IIb}$β$_3$ caused a significant increase in the number of leukocytes binding in a manner that could be blocked with GR144053 (*Figure 3d*).

The best characterized platelet-leukocyte interaction is mediated by P-selectin on activated platelets binding to PSGL-1 on leukocytes (*Vandendries et al., 2004*). Although we detected little/no P-selectin on the surface of VWF-'primed' platelets, this did not formally exclude a role for P-selectin in leukocyte adhesion. Therefore, we first established the efficacy of P-selectin blockade through the marked reduction of leukocyte binding to collagen captured/activated platelets (*Figure 3—figure supplement 1b–c*). However, blockade of P-selectin on FL-VWF-bound platelets from whole blood or plasma-free blood had no effect upon the number of leukocytes interacting with the platelet surface, suggesting that the recruitment of leukocytes is independent of P-selectin (*Figure 3e*). Leukocytes rolled faster over platelet surfaces after blocking P-selectin in plasma-free blood (*Figure 3f* and *Video 3*) or whole blood (*Figure 3—figure supplement 1d*), suggesting that whereas leukocyte capture is highly dependent on activated α$_{IIb}$β$_3$ (and not P-selectin), once recruited, small amounts of P-selectin on the platelet surface may slow leukocyte rolling.

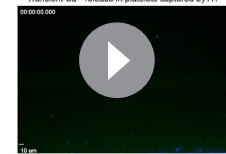

**Video 2.** VWF A1-GpIbα interaction induces intraplatelet Ca$^{2+}$ release under flow. Platelets were pre-loaded with Fluo-4 AM prior to perfusion through VWF A1*-coated microchannels at 1000 s$^{-1}$ for 5 min. Increases in platelet fluorescence corresponds to platelet intracellular Ca$^{2+}$ release following attachment to VWF A1*. Note the repeated transient increases in fluorescence under flow, indicative of sustained and repeated signaling stimuli.

https://elifesciences.org/articles/53353#video2

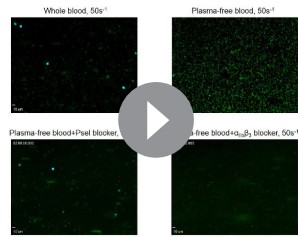

**Video 3.** Leukocytes bind/roll on VWF-'primed' platelets. Whole blood or plasma-free blood labeled with $DiOC_6$ was perfused through channels coated with VWF A1* at high shear for 3.5 min to capture platelets in the presence or absence of either a blocking anti-P-selectin antibody or eptifibatide ($\alpha_{IIb}\beta_3$ blocker) prior to the acquisition of videos. The shear rate was reduced to 50 s$^{-1}$ to visualize leukocyte rolling or attaching (tracked in blue). Leukocytes rolled/bound on platelets in whole blood, plasma-free blood and plasma-free blood containing anti-P-selectin blocking antibody, but not in the presence of eptifibatide ($\alpha_{IIb}\beta_3$ blocker). The use of the anti-P-selectin blocking antibody did, however, increase the rolling velocity of leukocytes over the platelet surface.

https://elifesciences.org/articles/53353#video3

## Leukocytes bind directly to activated $\alpha_{IIb}\beta_3$

To more specifically test the leukocyte interaction with activated $\alpha_{IIb}\beta_3$ (and to exclude other platelet receptors), purified $\alpha_{IIb}\beta_3$ was covalently coupled to microchannels and, thereafter, activated with Mn$^{2+}$ (*Litvinov et al., 2005*). Isolated peripheral blood mononuclear cells (PBMCs) and polymorphonuclear cells (PMNs) were perfused through $\alpha_{IIb}\beta_3$-coated channels at 50 s$^{-1}$. Cells from both PBMCs and PMNs (*Figure 4ai–ii & and b*) directly attached to the activated $\alpha_{IIb}\beta_3$ surface. This binding was significantly diminished (>70%) by adding either purified fibrinogen or eptifibatide to the PMNs (*Figure 4aiv-v and b*).

We also captured purified $\alpha_{IIb}\beta_3$ to flow channel surfaces using the activating anti-$\beta_3$ (LIBS) antibody. Leukocytes were again efficiently captured to this surface in a manner that could be inhibited (~70%) by blocking $\alpha_{IIb}\beta_3$ (*Figure 4b*).

Activated (rather than resting) leukocytes can interact with platelets via Mac-1 ($\alpha_M\beta_2$) (either directly through GPIbα or via fibrinogen bridge with activated $\alpha_{IIb}\beta_3$) or LFA-1 ($\alpha_L\beta_2$) via ICAM-2 (*Damle et al., 1992*; *Diacovo et al., 1994*; *Simon et al., 2000*; *Weber and Springer, 1997*). However, blocking $\beta_2$ suggested no role for either of these activated integrins in leukocyte binding to VWF-'primed' platelets (*Figure 4c*). In summary, we show leukocytes bind $\alpha_{IIb}\beta_3$ directly dependent upon its RGD-binding groove, but in a manner that is independent of Mac-1 or LFA-1.

## T-cells and neutrophils interact with VWF-'primed' platelets via activated $\alpha_{IIb}\beta_3$

We found no evidence of either CD14$^+$ monocytes or CD19$^+$ B-cells in PBMCs interacting with activated $\alpha_{IIb}\beta_3$. T-cells were the only cell type amongst the PBMCs capable of binding activated $\alpha_{IIb}\beta_3$ or VWF-'primed' platelets (*Figure 4d–e*).

Using isolated PMNs, we found that cells stained with anti-CD16 bound to activated $\alpha_{IIb}\beta_3$-coated channels and also to VWF-'primed' platelets (*Figure 4f and g*). Based on multi-lobulated segmented nuclear morphology (*Figure 4f*), these cells were indicative of CD16$^+$ neutrophils. Neutrophils scanned the platelet- or $\alpha_{IIb}\beta_3$-coated surfaces (*Figure 4g* and *Video 4*) suggesting that the binding of neutrophils to $\alpha_{IIb}\beta_3$ under flow may itself initiate signaling events within neutrophils. In line with this, PMNs bound to VWF-'primed' platelets (*Figure 5a*) or activated $\alpha_{IIb}\beta_3$ (*Figure 5b*) surfaces exhibited similar intracellular Ca$^{2+}$ release (*Video 5*) that reached a maximum after 200–300 s (*Figure 5c–d*).

## Binding of neutrophils to $\alpha_{IIb}\beta_3$ under flow induces Nox- and Ca$^{2+}$-dependent NETosis

Platelets assist in the targeting of intravascular bacterial pathogens through stimulation of the release of NETs (*Brinkmann et al., 2004*; *Gaertner et al., 2017*; *Wong et al., 2013*; *Yeaman, 2014*). However, the physiological agonists or mechanisms that drive NETosis are not fully resolved (*Nauseef and Kubes, 2016*). We therefore examined whether the binding of neutrophils to $\alpha_{IIb}\beta_3$ might induce NETosis. Isolated PMNs were perfused over either activated $\alpha_{IIb}\beta_3$ or anti-CD16 (negative control) at 50 s$^{-1}$ for 10 min and NETosis was subsequently analyzed under static conditions (*Figure 6a* and *Video 6*). Nuclear decondensation was evident after ~60 min, and Sytox Green fluorescence, indicative of cell permeability that precedes NETosis, was detected from ~85 min. Nuclear

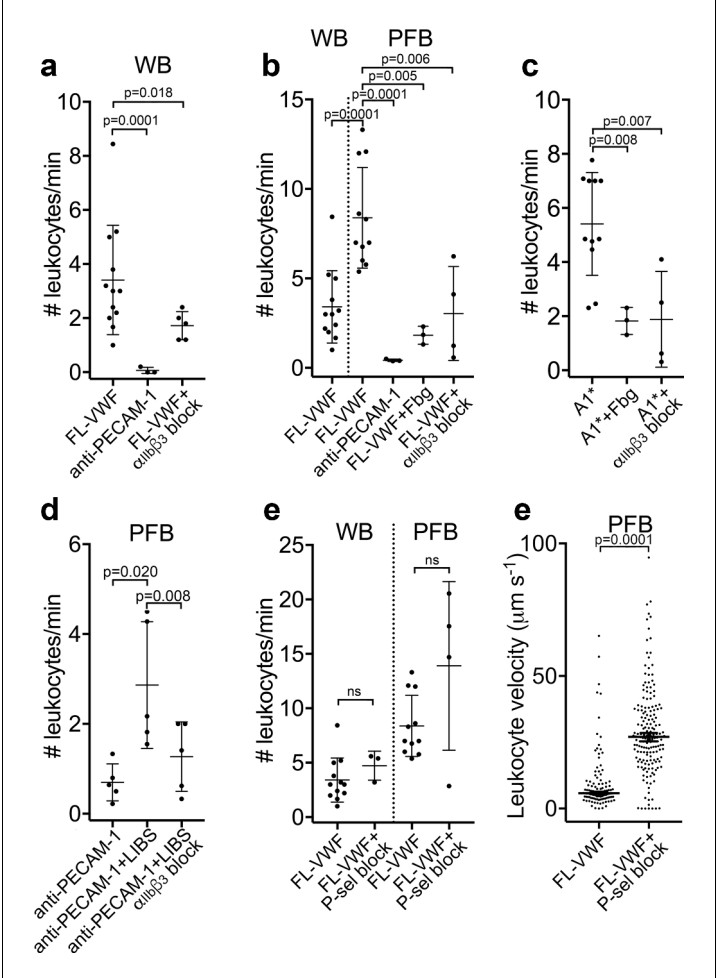

**Figure 3.** Leukocytes bind to VWF-bound platelets under flow (see also *Video 3*). (**a**) Graph of the number of leukocytes/minute in WB interacting with platelets bound to FL-VWF in the absence (n = 12) or presence of eptifibatide/GR144053 (n = 5) or binding to platelets bound to anti-PECAM-1 antibody (n = 3). (**b**) Graph of the number of leukocytes/minute in WB or PFB interacting with platelets bound to FL-VWF in the absence (n = 12) or presence of 1.3 mg/ml fibrinogen (n = 3) or eptifibatide/GR144053 (n = 4), or binding to platelets bound to anti-PECAM-1 antibody (n = 3). (**c**) Graph of the number of leukocytes/minute in PFB interacting with platelets bound to A1* in the absence (n = 11) or presence of fibrinogen (n = 3) or eptifibatide/GR144053 (n = 4). (**d**) Graph of the number of leukocytes/minute in PFB interacting with platelets bound to anti-PECAM-1 antibody in the absence (n = 5) or presence of LIBS/anti-$\beta_3$ activating antibody (n = 5)±GR144053 (n = 5). (**e**) Graph of the number of leukocytes/minute in WB or PFB, as shown, interacting with platelets bound to FL-VWF in the absence or presence of a blocking anti-P-selectin antibody. (**f**) Graph of leukocyte rolling velocity on platelets bound to FL-VWF in PFB in the absence or presence of a blocking anti-P-selectin antibody. Data shown are individual leukocyte rolling velocities (n=121 and 178, respectively) for three separate experiments. In all graphs, data plotted are mean ± SD. Data were analyzed using unpaired, two-tailed Student's t test; ns not significant.

The online version of this article includes the following source data and figure supplement(s) for figure 3:

**Source data 1.** Leukocytes bind to VWF-bound platelets source data.
**Figure supplement 1.** Antibody-mediated blockade of P-selectin diminishes leukocyte binding to collagen bound platelets.
**Figure supplement 1—source data 1.** P-selectin blocking source data.

decondensation, increased cell permeability and positive staining with a cell impermeable DNA fluorophore do not specifically identify NETosis. Therefore, an anti-citrullinated histone H3 antibody was perfused through the channels after 90 min to more specifically identify NETs. The introduction of flow at this point caused the DNA to form extended mesh-like NETs that were stained positively by

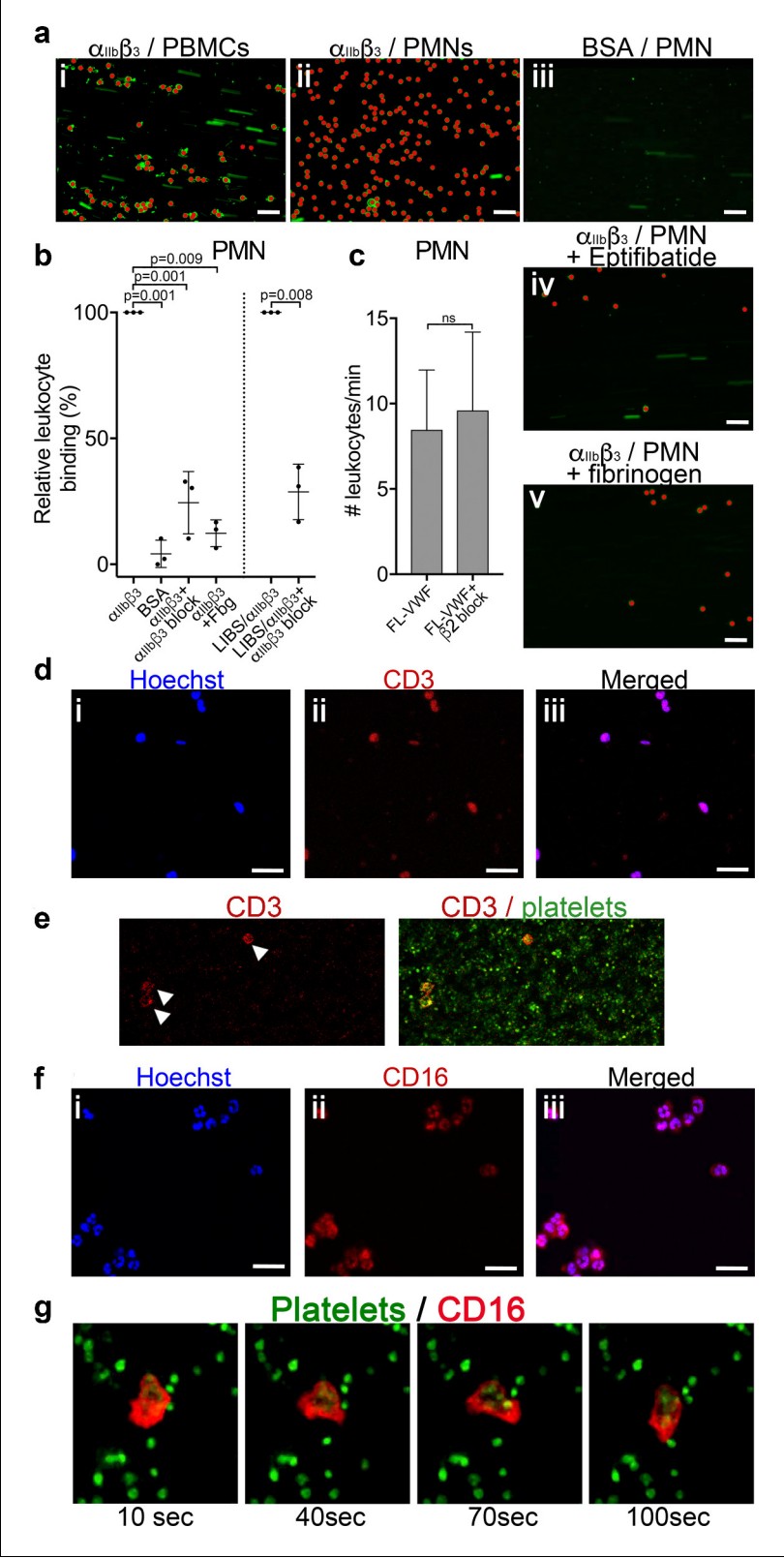

**Figure 4.** Leukocytes bind to activated $\alpha_{IIb}\beta_3$ under flow. (**a**) Purified $\alpha_{IIb}\beta_3$ or BSA, as noted, were covalently coupled to microchannel surfaces by amine coupling. $\alpha_{IIb}\beta_3$ was activated using $Mn^{2+}$ and $Ca^{2+}$ in all buffers. PBMCs (i) or PMNs (ii-v) labeled with $DiOC_6$ were perfused through channels at 50 $s^{-1}$ in the presence and absence of eptifibatide (iv) or 1.3 mg/ml purified fibrinogen (v). Bound leukocytes (as opposed to flowing) are

*Figure 4 continued on next page*

*Figure 4 continued*

pseudo-colored red to aid visualization and to distinguish from leukocytes in transit. Scale bar; 50 μm. (**b**) Graphical representation of relative leukocyte binding to activated $\alpha_{IIb}\beta_3$ in the presence and absence of eptifibatide or 1.3 mg/ml purified fibrinogen, or to BSA after 15 min of PMN perfusion (n = 3), or to $\alpha_{IIb}\beta_3$ captured and activated by LIBS/anti-$\beta_3$ activating antibody in the absence and presence of GR144053 (n = 3). Data plotted are mean ± SD. Data were analyzed using unpaired, two-tailed Student's t test. (**c**) Graph of the number of leukocytes/minute in PFB interacting with platelets bound to FL-VWF in the absence (n = 12) or presence of a blocking anti-β2 integrin polyclonal antibody (n = 5) capable of blocking both LFA-1 or Mac-1 on leukocytes. (**d**) PBMCs stained with Hoechst dye (i - blue), anti-CD3 (ii - red) and merged (iii). Representative of n = 4. Scale bar; 20 μm. (**e**) PFB stained with DiOC6 was perfused over FL-VWF at 1000 s$^{-1}$ followed by 50 s$^{-1}$. T-cells labeled with anti-CD3 (red - arrows) were seen to attach to 'primed' platelets **f**) PMNs stained with Hoechst dye (i - blue), anti-CD16 (ii - red) and merged (iii). Representative of n = 4. Scale bar; 20 μm (see also *Video 4*). (**g**) Images depicting a neutrophil stained with anti-CD16 (red) 'scanning' the 'primed' platelets stained with DiOC$_6$ (green). Images shown were taken 10, 40, 70 and 100 s after neutrophil attachment note the movement of the neutrophil shown - see also *Video 4*.

The online version of this article includes the following source data for figure 4:

**Source data 1.** Leukocytes bind to activated αIIbβ3 source data.

---

Hoechst and the anti-citrullinated histone H3 antibody (*Figure 6b*). Very similar results were obtained with PMN bound to either $\alpha_{IIb}\beta_3$ (captured by the activating anti-$\beta_3$ antibody), or to platelets 'primed' by A1* or FL-VWF, which all exhibited a similar proportion of neutrophils undergoing NETosis within the 2 hr timeframe, suggesting that interaction with $\alpha_{IIb}\beta_3$ alone can promote NETosis and that any platelet releasate present does not appreciably augment this process under these conditions. On activated $\alpha_{IIb}\beta_3$, 69% ± 14% of neutrophils through the entire channel formed NETs after 2 hr, compared to minimal NETosis events (8 ± 8%) when neutrophils were captured by anti-CD16 (*Figure 6c*). When neutrophils were captured on $\alpha_{IIb}\beta_3$ in the absence of flow, neutrophils attached, but NETosis was significantly reduced by fourfold, with only 17% of neutrophils exhibiting signs of NETosis (*Figure 6c*). This suggested that the signaling mechanism from the platelet to the neutrophil is mechano-sensitive and does not require other platelet receptors or releasate components.

Neutrophils captured by an anti-CD16 antibody and stimulated with phorbol 12-myristate 13-acetate (PMA) for 2 hr led to 100 ± 0.5% of neutrophils releasing NETs (*Figure 6d*). PMA-induced NETosis was not significantly inhibited in the presence of TMB-8 (an antagonist of intracellular Ca$^{2+}$ release; 90 ± 9%), but was effectively inhibited in by DPI (NADPH oxidase inhibitor; 16 ± 13%), similar to previous reports (*Gupta et al., 2014*). NETosis of neutrophils captured by $\alpha_{IIb}\beta_3$ under flow was significantly inhibited by TMB-8 (17 ± 9%) and DPI (25 ± 10%) (*Figure 6e*). This highlights the dependency of both intracellular Ca$^{2+}$ release and NADPH oxidase signaling pathways in NETosis in response to binding $\alpha_{IIb}\beta_3$ under flow.

## 'Primed' platelets interact with SLC44A2 receptor on neutrophils

Our data point to the presence of a specific receptor on the surface of neutrophils (and T-cells) that is not present on B cells or monocytes and that is capable of binding to activated $\alpha_{IIb}\beta_3$ and transducing a signal into the cell. To identify this leukocyte counter-receptor, we analyzed RNA sequencing data from different leukocyte populations, selecting genes that are expressed at higher levels in neutrophils (or in CD4$^+$ T-cells) than in monocytes (*Adams et al.,*

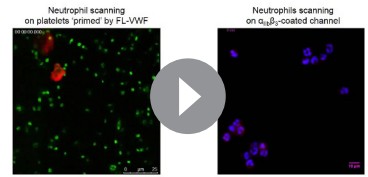

**Video 4.** Neutrophil start scanning following interaction with activated $\alpha_{IIb}\beta_3$. Following attachment of neutrophils (labeled with anti-CD16; red) to platelets (green) captured on FL-VWF (left), neutrophils start to scan the platelet surface. Similarly, neutrophils (anti-CD16-APC and Hoechst staining) bound to activated $\alpha_{IIb}\beta_3$-coated microchannels also scanned the surface and in some cases appeared to start migrating against the direction of flow. Shear rate = 50 s$^{-1}$.
https://elifesciences.org/articles/53353#video4

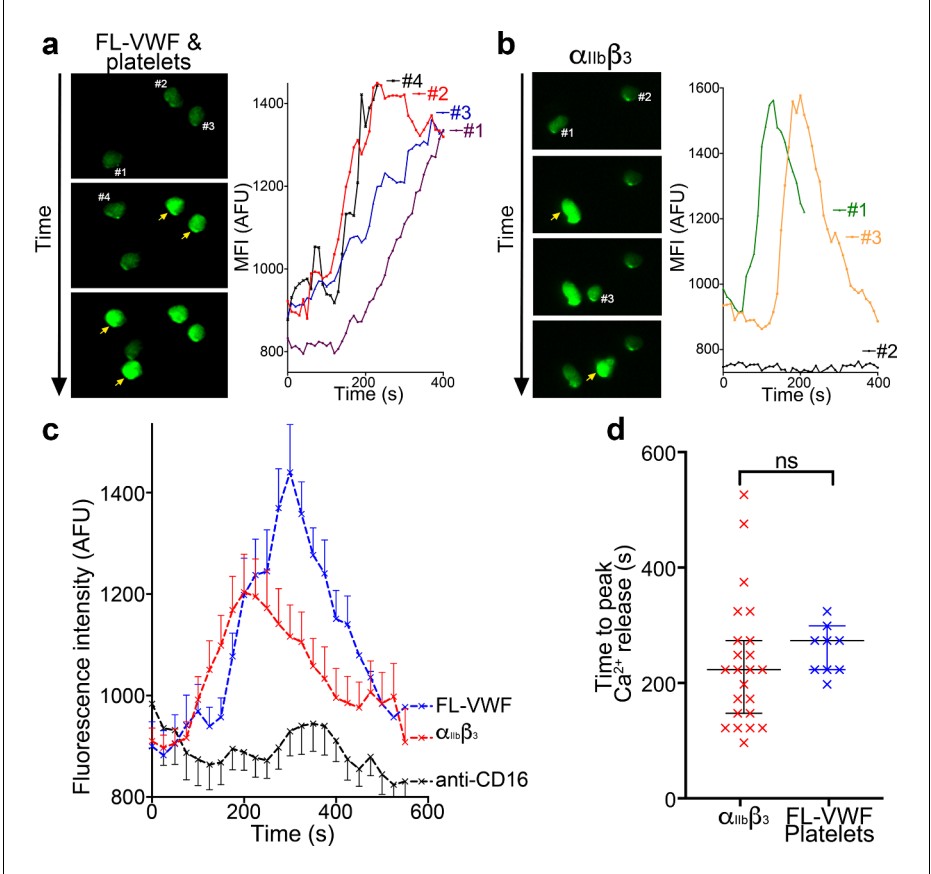

**Figure 5.** Binding to $\alpha_{IIb}\beta_3$ induces intracellular $Ca^{2+}$ release in neutrophils. (**a**) Representative images of neutrophils pre-loaded with Fluo-4 AM bound to VWF-'primed' platelets captured (*Video 5*). Neutrophils are numbered #1-#4. The yellow arrow highlights a frame in which the fluorescence has increased in the attached neutrophil. For each neutrophil shown, intracellular $Ca^{2+}$ release is quantified by measurement of cellular mean fluorescent intensity (MFI) over time. (**b**) As in A) except neutrophils were perfused over activated $\alpha_{IIb}\beta_3$. MFI increased for neutrophils #1 and #3, but not for neutrophil #2. (**c**) Graph depicting the change in MFI as a function of time after neutrophil attachment to microchannels coated with activated $\alpha_{IIb}\beta_3$ (n = 24 neutrophils from three different experiments), VWF-'primed' platelets (n = 9 neutrophils from one experiment) or anti-CD16 (n = 13 neutrophils from two different experiments). Data plotted are mean ± SEM. (**d**) Dot plot presenting the time between neutrophil attachment and maximum MFI of neutrophils binding to purified $\alpha_{IIb}\beta_3$ (red), or VWF-'primed' platelets (blue). Data plotted are median ±95% confidence interval. Data were analyzed using the Mann-Whitney test.

The online version of this article includes the following source data for figure 5:

**Source data 1.** Ca2+ release in neutrophils source data.

---

*2012*; *Grassi et al., 2019*; *Figure 7*). We further limited the candidate search by selecting those genes that code for transmembrane proteins. Using this approach, we identified 93 candidate genes. Of these, 33 genes were excluded as they are primarily associated with intracellular membranes. An additional 16 genes were also excluded due to the presence of short extracellular regions/domains (<30 a.a.) that would unlikely be capable of facilitating interactions with an extracellular binding partner (*Figure 7*). We then analyzed proteomic data to verify the preferential expression of the remaining candidates in neutrophils as opposed to monocytes (*Rieckmann et al., 2017*). These data suggested that the protein product of 14 of the remaining genes appeared to be detected in higher abundance in monocytes, which we used as a further exclusion criterion (*Figure 7—figure supplement 1a*). From the remaining 30 candidate genes, the *SLC44A2* gene was selected for validation due to its recent identification as a risk locus for both DVT and stroke (*Germain et al., 2015*; *Hinds et al., 2016*), both of which are pathologies associated with described

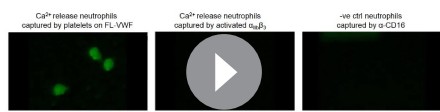

**Video 5.** Intracellular Ca$^{2+}$ release in neutrophils following interaction with activated $\alpha_{IIb}\beta_3$ under flow. Neutrophils were pre-loaded with Fluo-4 AM and perfused over FL-VWF 'primed' platelets, $\alpha_{IIb}\beta_3$-coated or anti-CD16 coated microchannels at 50 s$^{-1}$. Following attachment of neutrophils to VWF-'primed' platelets or activated $\alpha_{IIb}\beta_3$ (but not anti-CD16), an increase in fluorescence corresponding to neutrophil intracellular Ca$^{2+}$ release was observed.

https://elifesciences.org/articles/53353#video5

contributions of platelet-leukocyte interactions. SLC44A2 is a cell surface receptor with 10 membrane-spanning domains and five extracellular loops of 178a.a., 38a.a., 72a.a., 38a.a. and 18a.a. in length, respectively (*Nair et al., 2016*). We sourced antibodies against SLC44A2 that specifically recognize amino acid sequences within the first and second extracellular loops. Published proteomic profiling confirmed the preferential expression of SLC44A2 in neutrophils (*Figure 7—figure supplement 1a*; *Rieckmann et al., 2017*). Western blotting of isolated granulocyte lysates revealed two bands representing SLC44A2 (glycosylated and nascent/non-glycosylated SLC44A2) (*Figure 7—figure supplement 1b*). Perfusing human neutrophils over immobilized activated $\alpha_{IIb}\beta_3$ in the presence of the first anti-SLC44A2 antibody (anti-SLC44A2 #1) that recognizes the second extracellular loop revealed a dose-dependent blockade of neutrophil binding when compared to no antibody or control rabbit IgG (*Figure 8a–b*). A second anti-SLC44A2 antibody (anti-SLC44A2 #2) that recognizes the first extracellular loop region of SLC44A2 confirmed these findings (*Figure 8a*). The anti-SLC44A2 #2 almost completely blocked neutrophil binding to activated $\alpha_{IIb}\beta_3$ suggesting that this antibody more effectively blocks the neutrophil binding to the integrin than anti-SLC44A2 #1. This may suggest that the first and longest extracellular loop is involved in interaction with an extracellular ligand.

Based on these results, we transfected HEK293T cells with an expression vector for human SLC44A2 fused to turbo green fluorescent protein (tGFP) at the intracellular C-terminus. Transfected cells were perfused through activated $\alpha_{IIb}\beta_3$ coated channels and cell binding was quantified. Transfected cells bound to these surfaces in a manner that could be blocked by GR144053 (that blocks $\alpha_{IIb}\beta_3$) or by the anti-SLC44A2 #1 antibody (*Figure 8c*).

The SNP in *SLC44A2* identified by GWAS studies that is protective against VTE and stroke (rs2288904-A) causes a missense mutation (R154Q) in the first 178a.a. extracellular loop of SLC44A2 (*Germain et al., 2015*). Based on this, we hypothesized that this substitution might exert a functional influence upon the ability of SLC44A2 to interact with $\alpha_{IIb}\beta_3$. Consistent with this hypothesis, HEK293T cells transfected with the SLC44A2 (R154Q)-tGFP expression vector exhibited reduced ability to interact with immobilized $\alpha_{IIb}\beta_3$ (*Figure 8c*). Western blot analysis of transfected HEK293T cells revealed that expression of the SLC44A2(R154)-tGFP and SLC44A2(Q154)-tGFP was similar and that the tGFP remained uniformly associated with the fusion protein (*Figure 8c* inset).

To further explore the potential interaction between SLC44A2 and activated $\alpha_{IIb}\beta_3$ on platelets, we first captured and 'primed' platelets over VWF-coated surfaces and, thereafter, perfused SLC44A2-tGFP-transfected HEK293T cells. Again, these cells bound to VWF-'primed' platelets in a manner that could be blocked completely with GR144053 (to block $\alpha_{IIb}\beta_3$) or the anti-SLC44A2 #1 antibody (*Figure 8d–e*). Consistent with the previous results, HEK293T cells transfected with SLC44A2(R154Q) exhibited markedly reduced binding to VWF-'primed' platelets (*Figure 8d–e*). A previous report suggested that SLC44A2 might bind directly to VWF (*Bayat et al., 2015*). However, when SLC44A2-tGFP-transfected HEK293T cells were perfused of VWF surfaces, in the absence of platelets, no binding was detected (*Figure 8d*). Similarly, isolated neutrophils also failed to interact directly with VWF-coated surfaces, demonstrating the absolute dependence of platelets in facilitating cell capture under flow.

## Neutrophils homozygous for the rs2288904-A SNP exhibit reduced binding to activated $\alpha_{IIb}\beta_3$

The rs2288904-A SNP in *SLC44A2* has a minor allele frequency of 0.22 and is protective against VTE (*Germain et al., 2015*). It is therefore the common/wild-type allele, rs2288904-G, that is the risk

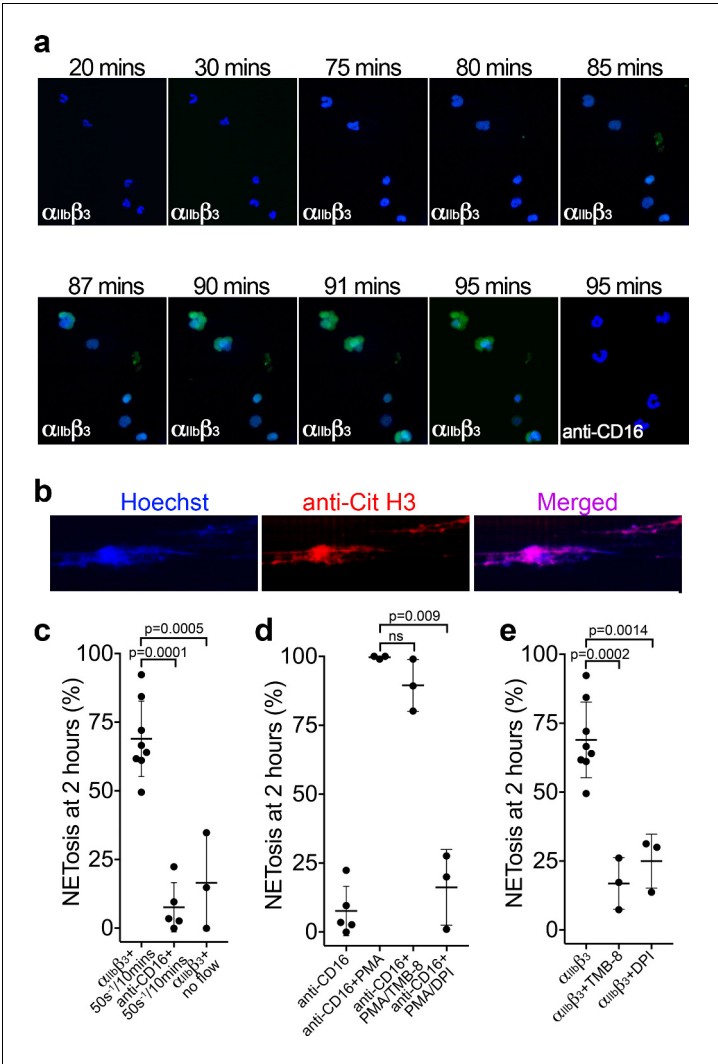

**Figure 6.** Binding of neutrophils to $\alpha_{IIb}\beta_3$ under flow induces NETosis. (**a**) Isolated PMNs labeled with Hoechst (blue) and cell-impermeable Sytox Green were perfused over $\alpha_{IIb}\beta_3$-coated microchannels, or anti-CD16, (-ve ctrl) at 50 s$^{-1}$ for 10 min and then monitored under static conditions. Representative composite images after 20, 30, 75, 80, 85, 87, 90, 91 and 95 min of attachment. Neutrophils bound to $\alpha_{IIb}\beta_3$ exhibited nuclear decondensation and increased cell permeability that precedes NETosis after about 85 min, Sytox Green staining appears, indicative of DNA becoming extracellular (see **Video 6**). Neutrophils bound to surfaces using an anti-CD16 antibody did not exhibit signs of NETosis or did so very rarely. (**b**) Immunostaining of neutrophils bound to $\alpha_{IIb}\beta_3$ after 90 mins as in a). Hoechst (blue), citrullinated H3 (red) and merged images are shown. (**c**) Graph showing the mean % of neutrophils ± SD in the entire microchannel that formed NETs after 2 hr of attachment on $\alpha_{IIb}\beta_3$ (n = 8) or anti-CD16 (n = 5), captured in the presence of flow (50 s$^{-1}$/10 min), or captured on $\alpha_{IIb}\beta_3$ under static/no flow conditions (n = 3). (**d**) Graph showing the mean % of neutrophils ± SD in the entire microchannel that formed NETs after 2 hr of attachment on anti-CD16 antibody in the presence of flow (50 s$^{-1}$/10 min) (n = 5) in the presence of PMA (n = 3), PMA and TMB-8 (n = 3) or PMA and DPI (n = 3). (**e**) Graph showing the mean % of neutrophils ± SD in the entire microchannel that formed NETs after 2 hr of attachment on $\alpha_{IIb}\beta_3$ in the presence of flow (50 s$^{-1}$/10 min) (n = 8) and in the presence of TMB-8 (n = 3) or DPI (n = 3), as noted. Data were analyzed using unpaired, two-tailed Student's t test; ns not significant.

The online version of this article includes the following source data for figure 6:

**Source data 1.** NETosis source data.

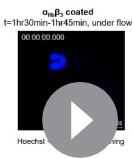

**Video 6.** $\alpha_{IIb}\beta_3$-induced NETosis. Isolated PMNs labeled with Hoechst (blue) and cell-impermeable Sytox Green were perfused over $\alpha_{IIb}\beta_3$-coated microchannels at 50 s$^{-1}$ for 10 min and then monitored under static conditions. Video shown represents period from 80 to 95 min after attachment. Sytox Green staining appears, indicative of DNA becoming extracellular highlights the neutrophils undergoing NETosis.

https://elifesciences.org/articles/53353#video6

allele for VTE with an odds ratio of 1.2–1.3. The frequency of individuals homozygous for the protective rs2288904-A allele amongst VTE cases is 30–50% lower than in healthy controls, which perhaps provides a better indication to its protective phenotype. Given its prevalence, we genotyped a group of healthy volunteers to identify individuals homozygous for the major allele (rs2288904-G/G), *SLC44A2* (R154/R154), and for the protective allele (rs2288904-A/A), *SLC44A2* (Q154/Q154) (*Figure 7—figure supplement 1c*). *SLC44A2* (R154/R154) neutrophils interacted with VWF-'primed' platelets as before (*Figure 8f* and *Video 7*). Consistent with the previous blocking experiments, this binding was partially blocked with anti-SLC44A2 #1, and almost completely blocked by anti-SLC44A2 #2 (*Figure 8f* and *Video 7*). Furthermore, and consistent with the transfection studies, neutrophils homozygous for the protective allele, *SLC44A2* (Q154/Q154), exhibited markedly reduced (~75%) binding to VWF-'primed' platelets (*Figure 8f* and *Video 7*) demonstrating a functional consequence of the rs2288904-A polymorphism on this neutrophil-platelet interaction. Consistent with this, neutrophils heterozygous form the polymorphism *SLC44A2* (R154/Q154) exhibited a trend toward intermediate binding to platelets (*Figure 8f*), although this did not reach statistical significance when compared to the *SLC44A2* (R154/R154) (p=0.07) or *SLC44A2* (Q154/Q154) (p=0.69) genotypes.

## Discussion

Although the ability of platelet GPIbα binding to VWF to mediate intraplatelet signaling events has been known for many years, the role that this signaling fulfils remains poorly understood (*Goto et al., 1995*). We demonstrate that under flow GPIbα-A1 binding 'primes', rather than activates, platelets, based on the rapid activation of $\alpha_{IIb}\beta_3$, but the lack of appreciable surface P-selectin exposure (*Figure 2* and *Figure 2—figure supplement 1b*). Some studies have reported that GPIbα-VWF-mediated signaling can induce modest α-granule release. However, the use of static conditions and processing of platelets may explain those observations. Despite this, when compared to other platelet agonists, degranulation and P-selectin exposure induced by VWF binding are both very low (*de Witt et al., 2014*; *Deng et al., 2016*).

That platelet binding to VWF under flow 'primes', rather than activates, platelets is consistent with in vivo observations. At sites of vessel damage, VWF is important for platelet accumulation through all layers of the hemostatic plug (*Joglekar et al., 2013*; *Lei et al., 2014*; *Verhenne et al., 2015*). All platelets within a thrombus/hemostatic plug likely form interactions with VWF. Despite this, it is only the platelets in the 'core' of the thrombus that become P-selectin-positive, procoagulant platelets, whereas the more loosely bound platelets that form the surrounding 'shell' remain essentially P-selectin-negative (*Welsh et al., 2014*). If VWF-binding alone were sufficient to fully activate platelets, the differential platelet characteristics of the 'core' and 'shell' would not be observed.

Although it is frequently implied that VWF is only important for platelet capture under high shear conditions, murine models of venous thrombosis with no collagen exposure have repeatedly revealed an important role for VWF-mediated platelet accumulation (*Bergmeier et al., 2008*; *Brill et al., 2011*; *Chauhan et al., 2007*). Platelet binding to VWF occurs most efficiently at arterial shear rates, but still occurs under lower linear venous shear (*Miyata and Ruggeri, 1999*; *Yago et al., 2008*; *Zheng et al., 2015*). However, linear channels do not mimic the distorted and branched paths of the vascular system that cause more disturbed flow patterns, particularly around valves. Using channels with changing geometry under lower shear conditions, we and others have noted that VWF captures platelets appreciably more efficiently in areas of disturbed flow (*Zheng et al., 2015*).

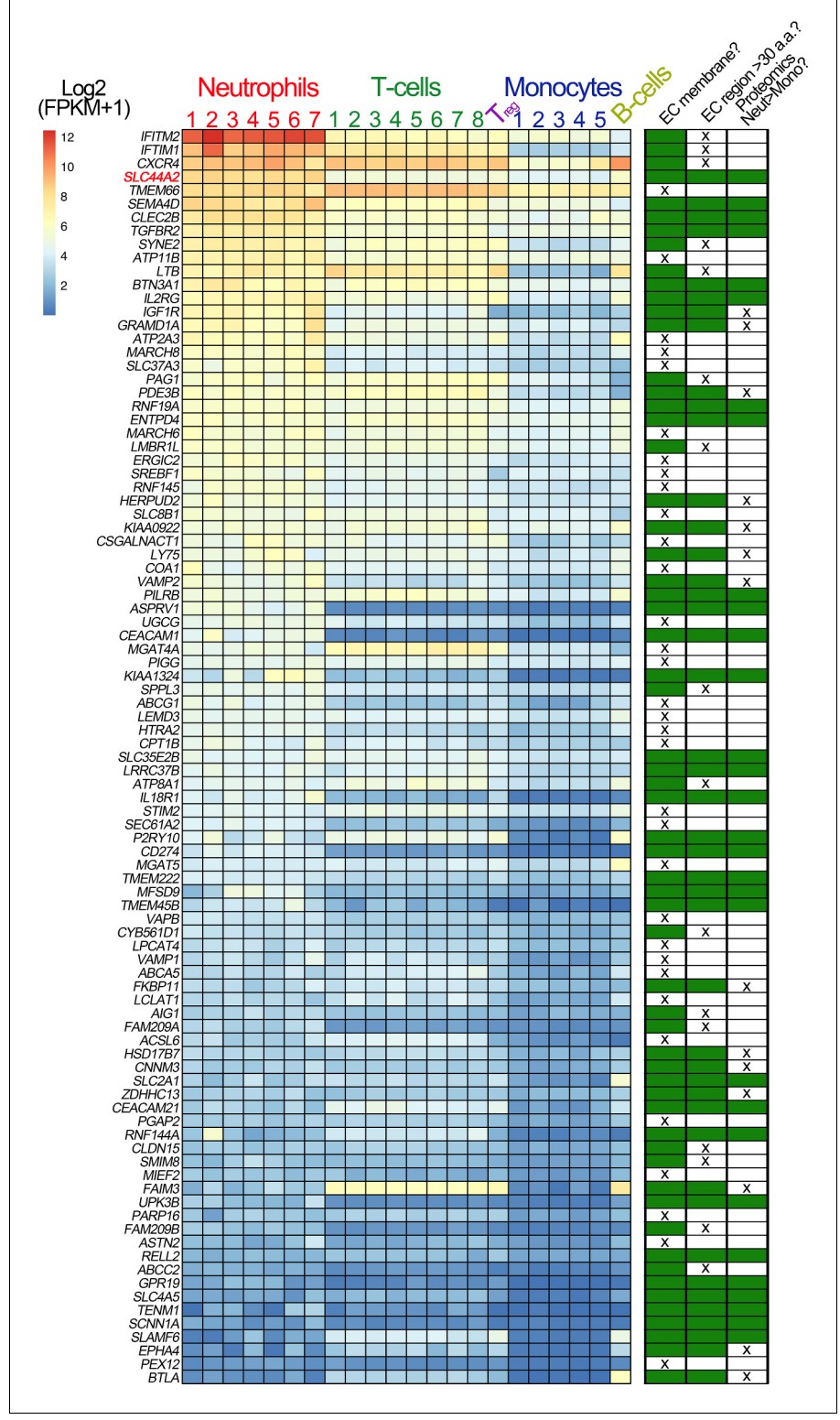

**Figure 7.** Transcriptomic profiling of human leukocytes. RNA sequencing data from different leukocytes were obtained from the BLUEPRINT consortium (*Grassi et al., 2019*). Differential gene expression analyses were performed: mature neutrophils (n = 7) vs monocytes (n = 5) and CD4-positive/αβ T cells (n = 8) vs monocytes (n = 5). Regulatory T cells (T$_{reg}$, n = 1) and native B cells (n = 1), are included in the heatmap, for comparison but were not used in differential gene expression analysis due to the low number of biological replicates. We first selected genes that were expressed significantly higher in neutrophils than in monocytes, and also those that were

*Figure 7 continued on next page*

*Figure 7 continued*

significantly higher in CD4-positive/αβ T cells than in monocytes. Their intersection identified 750 genes (598 of which protein coding). From these 598 genes, we selected the 93 genes that contained the Uniprot annotation of 'INTRAMEMBRANE DOMAIN' or 'TRANSMEM DOMAIN'. The effective log2(FPKM+1) data are presented in the heatmap of the 93 genes, with the rows ordered according to the mean neutrophil expression levels. Next to the heatmap is a table highlighting the subsequent selection criteria used to further narrow the search for candidate receptors for $\alpha_{IIb}\beta_3$. The first round of selection involved discarding those transmembrane proteins that are not present on the extracellular membrane, or primarily associated with intracellular membranes. The second selection criterion was to discard those proteins that had extracellular regions of <30 amino acids that might be less likely capable of mediating specific ligand binding. Finally, analysis of proteomic data from the ImmProt (http://immprot.org) resource was used to verify higher levels of protein of each selected gene in neutrophils than in monocytes.

The online version of this article includes the following source data and figure supplement(s) for figure 7:

**Figure supplement 1.** SLC44A2 expression in neutrophils and *SLC44A2* genotype analysis.
**Figure supplement 1—source data 1.** SLC44A2 Western blot source data.

---

Indeed, at venous flow rates through bifurcated channels (*Figure 9a*), we detected platelet capture on FL-VWF with concomitant 'priming' and leukocyte binding (*Figure 9b*). This was appreciably augmented at bifurcation points where disturbed flow exists. Consistent with our earlier findings, leukocyte binding was almost completely inhibited when $\alpha_{IIb}\beta_3$ was blocked (*Figure 9c*). This implies that VWF can function in platelet recruitment within the venous system, particularly in areas of turbulence (e.g. branch sites, valves), which are frequently the nidus for thrombus formation in DVT.

In venous thrombosis, the thrombus generally forms over the intact endothelium, in the absence of vessel damage. This poses the question of how VWF might contribute to DVT if subendothelial collagen is not exposed. It is likely that this reflects the function of newly secreted ultra-large VWF released from endothelial cells. Under low disturbed flow, released ultra-large VWF may tangle to form strings/cables over the surface of the endothelium. Tangled VWF strings/cables are appreciably more resistant to ADAMTS13 proteolysis than VWF that is simply unraveled. In the murine stenosis model of DVT, complete VWF-deficiency prevents platelet binding over the endothelium (*Bergmeier et al., 2008*; *Brill et al., 2011*; *Chauhan et al., 2007*). Similarly, blocking GPIbα binding to VWF also completely blocks platelet accumulation and thrombus formation in the stenosis model of DVT. Thus, when platelets bind to VWF under flow in such settings, platelets may become 'primed' facilitating both aggregation and neutrophil binding through activated $\alpha_{IIb}\beta_3$, but without activating them into procoagulant platelets.

Our study reveals, that T-cells and neutrophils can bind directly to activated $\alpha_{IIb}\beta_3$ on platelets or that has been coupled to microchannel surfaces (*Figure 3a–d* & *Figure 4a–b*). In both cases, the interaction can be inhibited by eptifibatide and GR144053 suggesting that both cell types may share the same receptor, in a manner that is dependent upon the RGD binding groove of activated $\alpha_{IIb}\beta_3$.

Previous studies have identified roles for $\beta_2$ integrins, Mac-1 ($\alpha_M\beta_2$) and LFA-1 ($\alpha_L\beta_2$), on leukocytes in mediating interactions with platelets. It should be recognized that the interactions of these molecules are dependent upon the integrins first being activated (and therefore also the cell), which is not the case in our system and is in contrast to previous studies implicating Mac-1 and LFA-1. However, we provide evidence that Mac-1 ($\alpha_M\beta_2$) and LFA-1 ($\alpha_L\beta_2$) are not involved by: 1) Leukocytes do not bind to 'unprimed' platelets captured by anti-PECAM-1. As Mac-1 and LFA-1 bind to GPIbα and ICAM-2, respectively, both of which are constitutively presented on the platelet surface (*Kuijper et al., 1998*; *Simon et al., 2000*), if Mac-1 and LFA-1 were the receptors involved binding would have been observed in these experiments (*Figure 3a–b*). 2) Mac-1 on leukocytes can bind indirectly to activated $\alpha_{IIb}\beta_3$ via a fibrinogen bridge (*Weber and Springer, 1997*). However, we demonstrate that fibrinogen competes for leukocyte binding to bind to activated $\alpha_{IIb}\beta_3$ (*Figure 3b–c* & *Figure 4b*). If fibrinogen were required, removal of fibrinogen from our perfusion system would have diminished the number of leukocyte interactions with primed platelets/$\alpha_{IIb}\beta_3$ if Mac-1 were involved. 3) Antibody-mediated blocking of $\beta_2$ integrins did not reduce VWF-'primed' platelet-leukocyte interactions (*Figure 4c*). 4) Only neutrophils and T cells interact with the 'primed' platelets, whereas Mac-1 and LFA-1 are also highly expressed in monocytes which do not bind (*Figure 4d–f*).

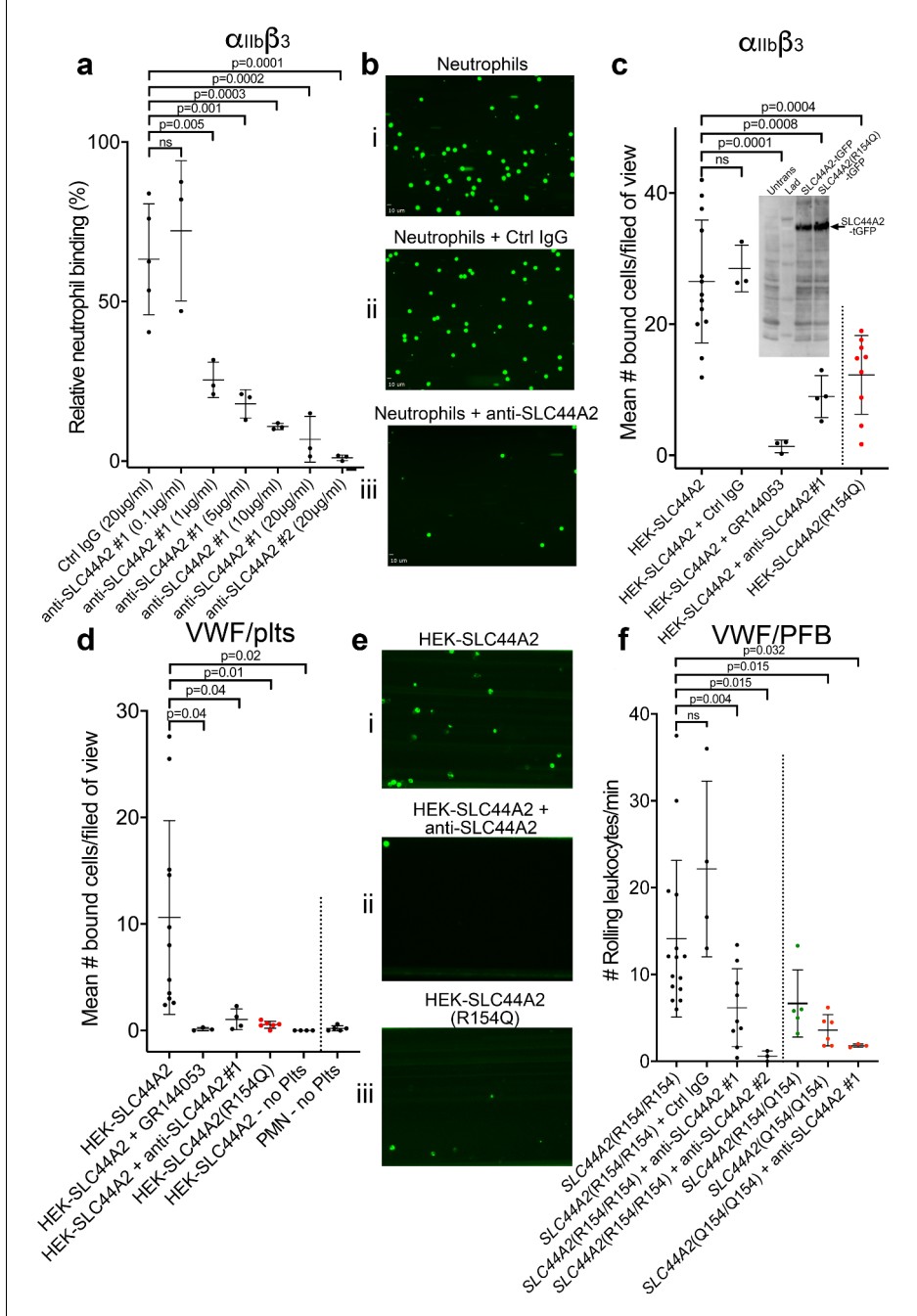

**Figure 8.** SLC44A2 binds activated $\alpha_{IIb}\beta_3$. (a) Graphical representation of relative neutrophil binding after 15 min of leukocyte perfusion at 50 s$^{-1}$ to activated $\alpha_{IIb}\beta_3$ captured and activated by LIBS2/anti-$\beta_3$ activating antibody in the presence and absence of increasing concentrations of anti-SLC44A2#1 or #2 antibodies. Data plotted are mean ± SD. Data were analyzed using One-Way ANOVA with multiple comparisons. (b) Representative micrographs of neutrophils bound to activated $\alpha_{IIb}\beta_3$ i) in the absence of antibody, ii) in the presence of control IgG and iii) in the presence of anti-SLC44A2#1. (c) Graphical representation of the number of HEK293T cells transfected with either SLC44A2-tGFP or SLC44A2(R154Q)-tGFP (shown in red) expression vectors binding to activated $\alpha_{IIb}\beta_3$ (captured and activated by LIBS2/anti-$\beta_3$ activating antibody) after 10 min flow at 25 s$^{-1}$. Experiments were performed in the presence and absence of either GR144053 or anti-SLC44A2#1 antibody. Data presented are the mean number of bound cells per field of view. Data were analyzed using one-way ANOVA with multiple comparisons. Inset shows a western blot of untransfected HEK293T cells, or HEK293T cells transfected with either SLC44A2-tGFP or SLC44A2(R154Q)-tGFP and detected using an anti-tGFP mAb. Only tGFP (26 kDa)

*Figure 8 continued on next page*

*Figure 8 continued*

fused to SLC44A2 (70 kDa) was detected in transfected cells; tGFP was uniformly associated with SLC44A2 and the SLC44A2(R154Q) variant (d) Graphical representation of the number of HEK293T cells transfected with either SLC44A2-tGFP or SLC44A2(R154Q)-tGFP (shown in red) expression vectors interacting with VWF-'primed' platelets. Plasma-free blood was first perfused at 1000 s$^{-1}$ for 3.5 mins to capture and 'prime' the platelets, and transfected HEK293T cells were subsequently perfused at 25 s$^{-1}$ for 10 min, in the presence and absence of GR144053 or anti-SLC44A2#1 antibody. HEK293T cells transfected with SLC44A2-tGFP or isolated neutrophils were also perfused over VWF in the absence of platelets, for 30 mins at 25 s$^{-1}$ or 50 s$^{-1}$ respectively. Data presented are the mean number of bound cells per field of view. Data were analyzed using one-way ANOVA with multiple comparisons. (e) Representative micrographs of HEK293T cells transfected with i) SLC44A2-tGFP in the absence of antibody, ii) SLC44A2-tGFP in the presence of anti-SLC44A2#1 antibody and iii) SLC44A2(R154Q)-tGFP bound to activated VWF-'primed' platelets. (f) Graphical representation of the number of neutrophils rolling per minute on VWF-'primed' platelets. PFB from individuals homozygous for the R154-encoding allele of *SLC44A2*(R154/R154) or the Q154-encoding allele of *SLC44A2*(Q154/Q154) (shown in red), or heterozygotes *SLC44A2*(R154/Q154) (shown in green) were perfused over 'primed' platelets for 10 min at 50 s$^{-1}$ (see *Video 7*). Experiments were performed in the presence and absence of anti-SLC44A2#1 or anti-SLC44A2#2 antibodies. Data plotted are mean ± SD. Data were analyzed using one-way ANOVA with multiple comparisons.

The online version of this article includes the following source data for figure 8:

**Source data 1.** SLC44A2 binds activated αIIbβ3 source data.

---

We also excluded a role for P-selectin-PSGL-1 for the platelet-leukocyte interaction that we observe as; 1) we detected little/no P-selectin on VWF-'primed' platelets (*Figure 2—figure supplement 1*), suggestive of minimal degranulation occurring; this also provides indirect evidence for the lack of CD40L on the platelet surface. 2) P-selectin blockade had no effect upon the number of leukocytes binding (*Figure 3e*), and 3) only T-cells and neutrophils bind VWF-'primed' platelets (*Figure 4*) - given that all leukocytes express PSGL-1,(*Laszik et al., 1996*) and CD40, if the capture of leukocytes were entirely P-selectin or CD40L-mediated, such cell-type selectivity would not be observed. We did, however, measure an influence of P-selectin upon the rolling speed of leukocytes over VWF-primed platelets (*Figure 3f* and *Figure 3—figure supplement 1d*). This suggests that although low levels of P-selectin present on the platelet surface is insufficient to facilitate leukocyte capture, it may synergize to slow rolling of leukocytes that are first captured by α$_{IIb}$β$_3$.

There are several studies that provide support for P-selectin-independent interactions of neutrophils and T-cells with platelets. Guidotti et al demonstrated the interaction of T-cells with small intrasinusoidal platelet aggregates in the liver during hepatotropic viral infections was independent of both P-selectin and CD40L in platelets (*Guidotti et al., 2015*). Using a murine model of peritonitis, Petri et al demonstrated that neutrophil recruitment and extravasation was highly dependent upon VWF, GPIbα, and platelets, but largely independent of P-selectin (*Petri et al., 2010*). Two further studies also corroborate the contention that VWF/GPIbα-bound platelets are capable of promoting neutrophil recruitment/extravasation in murine models of ischemia/reperfusion via P-selectin-independent mechanisms (*Gandhi et al., 2012*; *Khan et al., 2012*). These studies support the idea that both VWF and platelets can function beyond

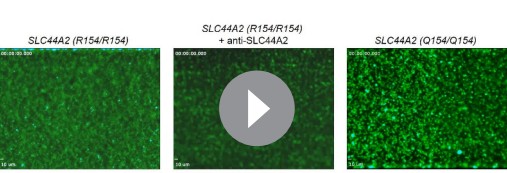

**Video 7.** Neutrophil binding to VWF-'primed' platelets occurs via SLC44A2 is modified by the rs2288904 SNP. Plasma-free blood generated from individuals homozygous for the rs2288904-G major allele in *SLC44A2* (SLC44A2 (R154/R154)) or the rs2288904-A minor allele (SLC44A2 (Q154/Q154)) SNP in *SLC44A2* was labeled with DiOC$_6$ and perfused through channels coated with VWF at high shear for 3.5 min. Shear was subsequently reduced to 50 s$^{-1}$. SLC44A2 (R154/R154) leukocytes (tracked in blue) were seen to roll on the VWF-'primed' platelets. The number of leukocytes interacting was severely reduced in the presence of the anti-SLC44A2 #2 antibody. Leukocytes homozygous for the rs2288904-A minor allele, SLC44A2 (Q154/Q154), associated with protection against venous thrombosis, exhibited reduced ability to interact with VWF-'primed' platelets.

https://elifesciences.org/articles/53353#video7

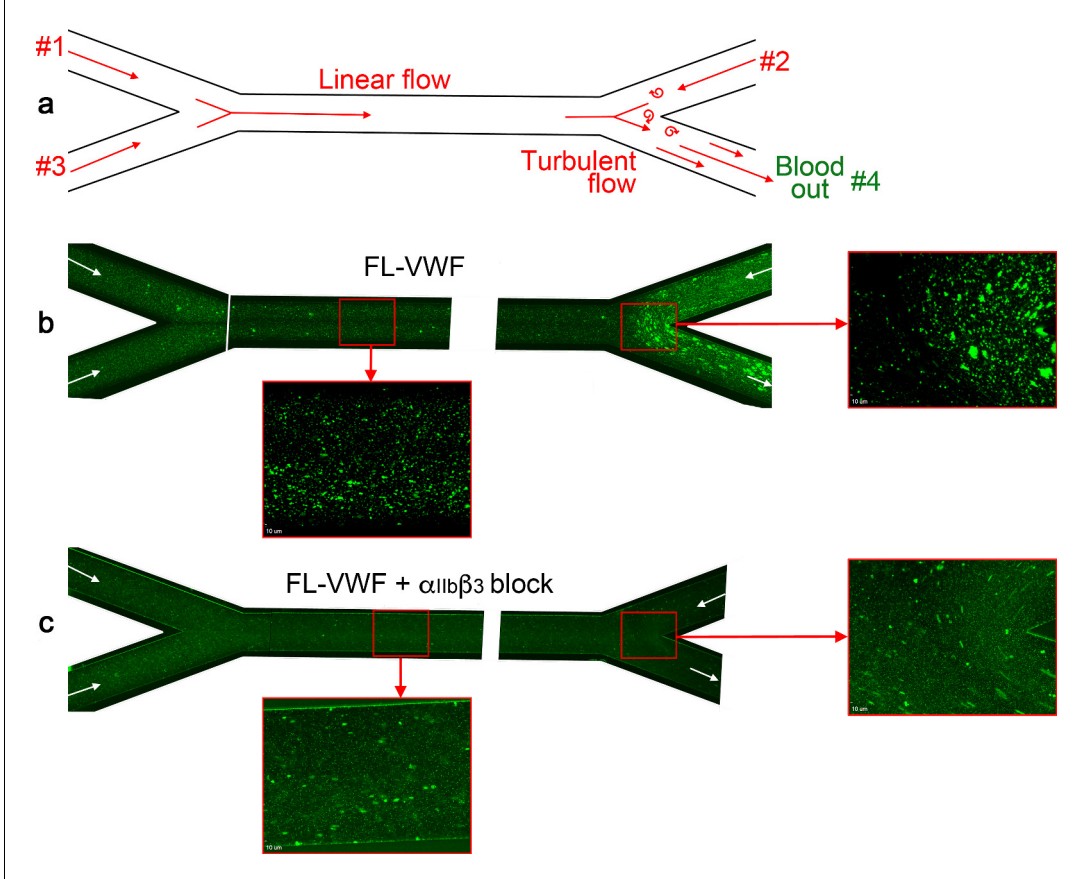

**Figure 9.** Analysis of platelet binding to FL-VWF under low/disturbed flow and subsequent leukocyte binding. (**a**) Schematic representation of blood flow through bifurcated channels. Blood is drawn through inlets #1, #2 and #3, and out through outlet #4. For much of the channels, the flow is linear, with particular exception to the bifurcation site on the right where disturbed flow exists due to convergence of flows. (**b**) Channels were coated with FL-VWF and whole blood labeled with DiOC6 was perfused through channels as in a), and as denoted by arrows, at an exit shear rate of $50~\text{s}^{-1}$. At this low shear rate, platelets can bind to the channel surface to which leukocytes (larger cells also stained in green) also bind. Although this is evident in the linear part of the channel (inset), at the site of most turbulent flow increased platelet and leukocyte binding was observed. (**c**) As in b) except GR144053 was added to block $\alpha_{IIb}\beta_3$. Blocking $\alpha_{IIb}\beta_3$ inhibited the majority of leukocyte binding to platelets (the majority of those observed are in transit). However, in the absence of leukocyte binding the binding of platelets to the VWF surface under low flow can be more clearly observed in both the linear and turbulent flow areas.

hemostasis to fulfil a role in leukocyte recruitment at sites of inflammation.

As T-cells and neutrophils (and not B-cells or monocytes) can bind platelets via activated $\alpha_{IIb}\beta_3$, this suggests that a specific receptor exists on these cells that is absent on B-cells or monocytes. Using transcriptomic and proteomic data, we identified 30 transmembrane candidates that were preferentially expressed in neutrophils (or T-cells) over monocytes. From this list, *SLC44A2* stood out due to its recent identification as a risk locus for both VTE and stroke, but with as yet unknown functional association with these pathologies (*Apipongrat et al., 2019*; *Germain et al., 2015*; *Hinds et al., 2016*). As platelet-leukocyte interactions are involved in both of these thrombotic disorders, we hypothesized that SLC44A2 functions as the neutrophil counter receptor for activated $\alpha_{IIb}\beta_3$. The cellular function of SLC44A2 is not well-defined. It contains 10 transmembrane domains with five extracellular loops. The intracellular N-terminal tail contains several putative phosphorylation sites of unknown functional significance. As well as neutrophils, SLC44A2 expression has also been reported in endothelial cells and platelets. However, proteomic data suggest that levels in neutrophils are >300 fold greater in neutrophils that platelets (*Rieckmann et al., 2017*).

We provide several lines of evidence to support the direct interaction between SLC44A2 and activated $\alpha_{IIb}\beta_3$. 1) Two different anti-SLC44A2 antibodies that recognize extracellular loops of the receptor blocked the binding of neutrophils to both VWF-'primed' platelets and to activated $\alpha_{IIb}\beta_3$.

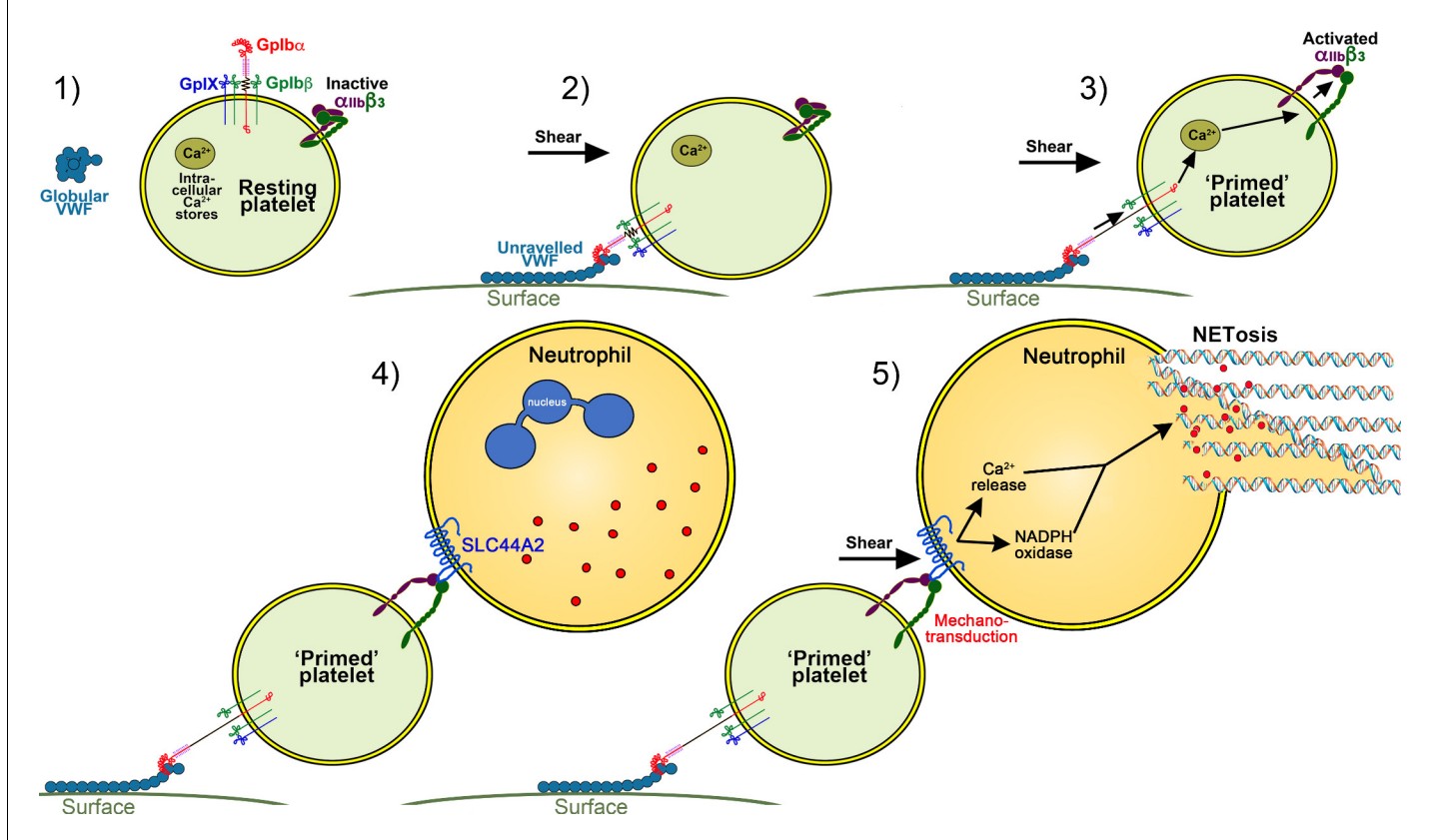

**Figure 10.** Model of platelet priming, neutrophil binding and NETosis. (1) Under normal conditions, VWF circulates in plasma in a globular conformation that does not interact with platelets. Resting platelets present GPIbα on their surface - in complex with GPIbβ, GPIX and GPV - and also $\alpha_{IIb}\beta_3$ in its inactive conformation. (2) When VWF is attached to a cell surface (e.g. activated endothelial cell/Kuppfer cell or to a bacterial cell) or to an exposed collagen surface under flow, VWF unravels to expose its A1 domain enabling capture of platelets via GPIbα. (3) Binding of platelets to VWF under flow induces mechano-unfolding of the juxtamembrane stalk of GPIbα leading to intraplatelet signaling, release of intraplatelet $Ca^{2+}$ stores and activation of integrin $\alpha_{IIb}\beta_{3.}$(4) Neutrophils can bind to activated $\alpha_{IIb}\beta_3$ under flow via SLC44A2. (5) Shear forces on the neutrophil induce mechanosensitive signaling into the neutrophil causing intracellular $Ca^{2+}$ release and NADPH oxidase-dependent NETosis.

2) Recombinant expression of SLC44A2 in HEK293T cells imparted the ability of these cells to bind both VWF-primed platelets and activated $\alpha_{IIb}\beta_3$ under flow in a manner that can be blocked by either GR144053 or by anti-SLC44A2 antibodies. 3) Introduction of the rs2288904-A SNP in *SLC44A2* that is protective against VTE resulted in markedly reduced binding of transfected HEK293T cells to both VWF-'primed' platelets and activated $\alpha_{IIb}\beta_3$. 4) Neutrophils homozygous for the rs2288904-A/A SNP exhibit significantly reduced binding to VWF-'primed' platelets.

Although NET production is an established mechanism through which neutrophils control pathogens (*Brinkmann et al., 2004*), many questions remain as to how NETosis is regulated (*Nauseef and Kubes, 2016*). Binding of platelets to Kupffer cells in the liver of mice following infection with *B. cereus* or *S. aureus* is mediated by VWF (*Wong et al., 2013*). This binding augments the recruitment of neutrophils, NET production and the control of infection (*Kolaczkowska et al., 2015*). Mice lacking VWF or GPIbα do not form these aggregates and so have diminished neutrophil recruitment and, therefore, decreased survival (*Wong et al., 2013*). How NETosis is initiated following platelet binding remains uncertain. Alone, lipopolysaccharide (LPS) is not a potent activator of NETosis (*Clark et al., 2007*). However, LPS-stimulated platelets, which bind of fibrinogen (i.e. $\alpha_{IIb}\beta_3$ is activated) and also robustly activate NETosis independent of P-selectin (*Clark et al., 2007*; *Looney et al., 2009*; *Lopes Pires et al., 2017*; *McDonald et al., 2012*).

We detected rapid release of intracellular $Ca^{2+}$ (within minutes) in bound neutrophils (*Figure 5* and *Video 5*) that preceded the release of NETs after 80–90 min (*Figure 6*). This process was dependent upon neutrophils being captured under flow, suggesting that signal transduction through

binding of $\alpha_{IIb}\beta_3$ to SLC44A2 may be mechanosensitive, which is consistent with the recent report suggesting a major influence of shear upon NETosis in the presence of platelets (*Yu et al., 2018*). As NETosis can be induced following binding to purified $\alpha_{IIb}\beta_3$ under flow alone, this suggests that this process does not require a component of the platelet releasate (e.g. high mobility group box 1, platelet factor 4, RANTES and thromboxane A2) which have been reported to be capable of driving NETosis (*Carestia et al., 2016*). Naturally, if platelet degranulation occurs (in response to other agonists), these components may have the potential to further augment this process.

We propose a model in which platelets have a tunable response that can distinguish their roles in hemostasis and immune cell activation (*Figure 10*). The 'priming' of platelets by binding to VWF under flow (in the absence of other platelet agonists) may assist in the targeting of leukocytes to resolve pathogens or mediate vascular inflammatory response. The activated $\alpha_{IIb}\beta_3$ integrin can then mediate neutrophil recruitment through binding to SLC44A2 (*Figure 10*). We do not exclude a supporting role for P-selectin in maintaining T-cell/neutrophil recruitment, but this is not essential for initiating recruitment. Under flow SLC44A2 transduces a mechanical stimulus capable of promoting NETosis via a pathway involving synergy between NADPH oxidase and $Ca^{2+}$ signaling (*Figure 10*). Homeostatically, this may be beneficial for immune responses. However, during chronic infection or vascular inflammation NET production may promote intravascular thrombosis. It remains to be determined whether the rs2288904-A SNP in *SLC44A2* that is protective against VTE might also (negatively) influence the host response to infection, or certain routes of infection that are more reliant upon platelet immune cell function. This should determine the net selective pressure upon the SNP. However, as VTE is a comparatively recent selective pathology (in evolutionary terms) and rates of VTE are typically low during the most common child-bearing years, this positive selective pressure on rs2288904-A SNP may not strong. Indeed, the prevalence of this SNP in different global populations suggests that the SNP has not originated recently. Moreover, if the host response to infection is, at least in certain settings, also impaired in individuals carrying the rs2288904-A SNP, this would be predicted to slow the rate of positive selection. More work is needed to understand role of rs2288904-A SNP in VTE and infection risk and penetrance of SNP within large genomic datasets.

This study identifies activated $\alpha_{IIb}\beta_3$ as a receptor and agonist for neutrophils through SLC44A2. This provides a previously uncharacterized mechanism of how platelet-neutrophil cross-talk is manifest in innate immunity; it also provides an explanation for how VWF and platelet-dependent neutrophil recruitment and NETosis may occur in thrombotic disorders such as DVT (*Laridan et al., 2019*), but also thrombotic microangiopathies like thrombotic thrombocytopenic purpura (*Fuchs et al., 2012b*). Identification of SLC44A2 as the counter-receptor for activated $\alpha_{IIb}\beta_3$ in conjunction with the prior identification of the protective rs2288904-A SNP in SLC44A2 that impairs the binding of neutrophils to platelets highlights SLC44A2 as a potential therapeutic target. Recent data reveal that SLC44A2-deficient mice exhibit normal hemostatic responses (*Tilburg et al., 2018*) but are protected against development of venous thrombosis (*Tilburg et al., 2020*), provide further encouragement for this strategy.

# Materials and methods

## Key resources table

| Reagent type (species) or resource | Designation | Source or reference | Identifiers | Additional information |
|---|---|---|---|---|
| Recombinant DNA reagent | *pMT-puro VWFA1-V5-His* | In house | | The insect cell expression vector to express VWF A1 domain can be obtained from Prof Crawley |
| Recombinant DNA reagent | *pMT-puro VWFA1*- V5-His* | In house | | The insect cell expression vector to express VWF A1* domain can be obtained from Prof Crawley |

*Continued on next page*

*Continued*

| Reagent type (species) or resource | Designation | Source or reference | Identifiers | Additional information |
|---|---|---|---|---|
| Bacterial strain, strain background (*E. coli*) | TOP10 (DH10B background) | Invitrogen | C4040-03 | Chemically competent cells |
| Antibody | anti-6xHis-HRP (*rabbit polyclonal*) | Abcam | RRID:AB_298652 | (WB: 1:10000) |
| Cell line (*D. melanogaster*) | S2 | S2 cells originally from ATCC transfected and selected in house | VWF A1-*V5-His*/S2 and VWF A1*-*V5-His*/S2 | Cell line maintained in Crawley lab; stably transfected with human VWF A1 domain-*V5-His* or VWF A1*-*V5-His* |
| Cell line (*Homo-sapiens*) | Human embryonic Kidney HEK293T | ATCC | RRID:CVCL_0063 | |
| Antibody | anti-VWF (*rabbit polyclonal*) | Agilent | RRID:AB_2315602 | (WB 1:1000) |
| Biological sample (*Homo sapiens*) | $\alpha_{IIb}\beta_3$ | Enzyme Research Laboratories | GP2b3a | $\alpha_{IIb}\beta_3$ isolated from human platelets |
| Antibody | anti-PECAM-1 (*mouse monoclonal*) | BioLegend | RRID:AB_314328 | (flow assays: 0.25 mg/ml IF:50 µg/ml) |
| Antibody | anti-beta3 (LIBS2) (*mouse monoclonal*) | Millipore | RRID:AB_10806476 | (flow assays 0.25 mg/ml) |
| Antibody | anti-CD16 (*mouse monoclonal*) | eBiosciences | RRID:AB_467129 | (flow assays 0.25 mg/ml) |
| Chemical compound, drug | eptifibatide | Sigma | SML1042-10MG | $\alpha_{IIb}\beta_3$ blocking/ inhibition 2.4 µM |
| Chemical compound, drug | GR144053 | Tocris | 1263 | $\alpha_{IIb}\beta_3$ blocking/ inhibition 2 µM |
| Antibody | anti-P-selectin (*mouse monoclonal*) | BD Biosciences | RRID:AB_395908 | Clone Ak4 (blocking P-selectin: 50 µg/ml) |
| Antibody | anti-SLC44A2 #1 (*rabbit polyclonal*) | Abcam | RRID:AB_2827953 | (SLC44A2 blocking 0–20 µg/ml) |
| Antibody | anti-SLC44A2 #2 (*rabbit polyclonal*) | LS Bio | RRID:AB_2827954 | (SLC44A2 blocking 0–20 µg/ml) |
| Antibody | anti-CD14-APC (*mouse monoclonal*) | BioLegend | RRID:AB_314190 | (IF 1:20) |
| Antibody | anti-CD3-APC (*mouse monoclonal*) | BioLegend | RRID:AB_314066 | (IF 1:20) |
| Antibody | anti-CD19-APC (*mouse monoclonal*) | BioLegend | RRID:AB_314242 | (IF 1:20) |
| Antibody | anti-CD16-APC (*mouse monoclonal*) | eBiosciences | RRID:AB_2016663 | (IF 1:20) |
| Chemical compound, drug | TMB-8 | Sigma | T111 | PKC inhibition: 20 µM |
| Chemical compound, drug | DPI | Sigma | 300260 | NADPH oxidase inhibition: 30 µM |
| Recombinant DNA reagent | pCMV6 Entry/SLC44A2-tGFP (plasmid with cDNA) | Origene | RG207181 | Expression vector of human SLC44A2 with C-terminal turbo GFP fusion |

*Continued on next page*

*Continued*

| Reagent type (species) or resource | Designation | Source or reference | Identifiers | Additional information |
|---|---|---|---|---|
| Recombinant DNA reagent | pCMV6 Entry/SLC44A2(R154Q)-tGFP (plasmid with cDNA) | In house | | As above but containing SLC44A2 (R154Q) variant. The expression vector pCMV6 Entry/SLC44A2(R154Q)-tGFP can be obtained from Prof Crawley |
| Sequence-based reagent | R154Q top primer | Invitrogen | | 5'-GTG GCT GAG GTG CTT CAA GAT GGT GAC TGC CCT-3' |
| Sequence-based reagent | R154Q bot primer | Invitrogen | | 5'-AGG GCA GTC ACC ATC TTG AAG CAC CTC AGC CAC-3' |
| Sequence-based reagent | SLC44A2 top sequencing primer | Invitrogen | | 5'-ACC TCA CGT ACC TGA ATG-3' |
| Sequence-based reagent | SLC44A2 bot sequencing primer | Invitrogen | | 5'-AGC CAT GCC CAT CCT CAT AG-3' |
| Antibody | anti-tGFP (*mouse monoclonal*) | Origene | RRID:AB_2622256 | (WB:0.3 µg/ml) |
| Antibody | anti-Cit H3 (*rabbit polyclonal*) | Abcam | RRID:AB_304752 | (IF:10 µg/ml) |
| Software | GraphPad Prism | GraphPad Prism | RRID:SCR_002798 | |
| Software | SlideBook 6.0 | 3i | RRID:SCR_014300 | |

## Preparation of VWF A1 domain and multimeric VWF

The coding sequence for the human VWF A1 domain (Glu1264 to Leu1469) was cloned into the pMT-puro vector, containing a C-terminal V5 and polyhistidine tag. The Y1271C/C1272R mutations were introduced by PCR into the A1 domain (A1*) (*Blenner et al., 2014*). All vectors were verified by sequencing.

S2 insect cells stably expressing either VWF A1-V5-His or VWF A1*-V5-His were selected using puromycin (Life Technologies). Cells were cultured under sterile conditions at 28°C in Schneider's-*Drosophila* medium (Lonza), supplemented with 10% heat-inactivated fetal bovine serum (FBS), 50 µg/ml penicillin and 50 U/ml streptomycin. Cells were grown in suspension in 2L conical flasks to a density of $2 \times 10^6$ cells/ml. Expression of VWF A1 or A1* was induced by addition of 500 µM $CuSO_4$ for 5–7 days, at 28°C and 110 rpm. Conditioned media from S2 cells were tested using the Venor GeM mycoplasma detection kit (Sigma) and confirmed to be mycoplasma free.

Conditioned media were harvested, cleared by centrifugation, concentrated by tangential flow filtration and dialyzed against 20 mM Tris (pH 7.8) 500 mM NaCl. VWF A1 or A1* were purified by a two-step purification method using a $Ni^{2+}$-HiTrap column followed by a heparin-Sepharose column (GE Healthcare) and elution with 20 mM Tris, 600 mM NaCl. VWF A1 and A1* were dialyzed in phosphate-buffered saline (PBS). A1 and A1* concentrations were determined by absorbance at 280 nm. Proteins were analyzed by SDS-PAGE under reducing and non-reducing, and by western blotting using anti-His (RRID:AB_298652) or anti-VWF (RRID:AB_2315602) antibodies. Full length, multimeric VWF was isolated from Haemate P by gel filtration and quantified by a specific VWF ELISA, as previously described (*O'Donnell et al., 2005*).

## Blood collection and processing

Fresh blood was collected from consented healthy volunteers in 40 µM PPACK (for whole blood experiments), 3.13% citrate (for leukocyte isolation) or 85 mM sodium citrate, 65 mM citric acid, 111 mM D(+) glucose, pH 4.5 (1x ACD, for plasma-free blood preparation). For reconstituted plasma-free blood, red blood cells (RBCs) and leukocytes were pelleted and washed twice. Separately, platelets were washed twice in 1x HEPES-Tyrode (HT) buffer containing 0.35% BSA, 75mU apyrase and 100 nM prostaglandin E1 (Sigma). RBCs, leukocytes and platelets were resuspended in 1x HT buffer supplemented with 0.35% BSA. In some experiments, 1.3 mg/ml purified fibrinogen (Haem Tech)

was added. For Ca$^{2+}$ assays, PRP was incubated with 5 µM Fluo-4 AM (Thermo Fisher Scientific) for 30 min at 37°C prior to washing, and plasma-free blood was recalcified with 1 mM CaCl$_2$ (final concentration) immediately prior to flow experiments.

PMNs and PBMCs separated using Histopaque1077 and Histopaque1119 were resuspended in 1x HT, supplemented with 1.5 mM CaCl$_2$. For Ca$^{2+}$ assays, PMNs were preloaded with 1 µM Fluo-4 AM for 30 min at 37°C, before washing. This study was approved by the Imperial College Research Ethics Committee (approval reference 19IC5523), and informed consent and consent to publish was obtained from all healthy volunteers.

## Flow experiments

VenaFluoro8+ microchips (Cellix) were coated directly with 2 µM VWF in PBS overnight at 4°C in a humidified chamber. Coated channels were blocked for 1 hr with 1x HEPES-Tyrode (HT) buffer containing 1% bovine serum albumin (BSA). For the isolated VWF A1 and A1* domains, NTA PEGylated microchips (Cellix) were used to capture the A1 or A1* via their His tags (*Tischer et al., 2014*). Channels were stripped with EDTA before application of Co$^{2+}$ and washing with 20 mM HEPES, 150 mM NaCl, pH 7.4 (HBS). To each channel, 20 µl of 3.75 µM VWF A1 or A1* were applied at room temperature for 20 min in a humidified chamber. Channels were then incubated with H$_2$O$_2$ for 30 min to oxidize Co$^{2+}$ to Co$^{3+}$, which stabilizes the binding of His-tagged A1/A1* (*Wegner et al., 2016*).

To NHS-microchannels (Cellix), 2.6 µM purified $\alpha_{IIb}\beta_3$ (ERL), 0.25 mg/ml PECAM-1 (RRID:AB_ 314328), anti-$\beta_3$/LIBS2 antibody (RRID:AB_10806476), anti-CD16 (RRID:AB_467129) or 0.25 mg/ml BSA were covalently attached by amine-coupling according to manufacturer's instructions. For directly coated $\alpha_{IIb}\beta_3$ channels, the surface was washed with HBS containing 1 mM MnCl$_2$, 0.1 mM CaCl$_2$ following coating. Mn$^{2+}$ was maintained in all subsequent buffers to cause $\alpha_{IIb}\beta_3$ to favor its open, ligand binding conformation, as previously reported (*Litvinov et al., 2005*).

To anti-$\beta_3$/LIBS2 antibody-coated channels, $\alpha_{IIb}\beta_3$ (ERL) was perfused over the surface to facilitate both capture and activation of $\alpha_{IIb}\beta_3$ on the surface.

Whole blood or plasma-free blood was perfused through channels coated with either FL-VWF, A1, A1* or anti-PECAM-1 at shear rates of 500–1500 s$^{-1}$ for 3.5 min, followed by 50 s$^{-1}$ for 15 min using a Mirus Evo Nanopump and Venaflux64 software (Cellix). In separate experiments, 2.4 µM eptifibatide (Sigma), 2 µM GR144053 (Tocris), or 50 µg/ml anti-P-selectin blocking antibody (RRID: AB_395908) were supplemented to whole blood or plasma-free blood. DiOC$_6$ (2.5 µM; Invitrogen) was used to label platelets and leukocytes. Cells were monitored in real-time using an inverted fluorescent microscope (Zeiss) or a SP5 confocal microscope (Leica). Leukocytes and platelets were distinguished by their larger size. For presentation and counting purposes, leukocytes were pseudo-colored to distinguish them. In some experiments, antibodies that recognize the second extracellular loop of SLC44A2, rabbit anti-SLC44A2 #1 (RRID:AB_2827953) or the first extracellular loop, rabbit anti-SLC44A2 #2 (RRID:AB_2827954) (0–20 µg ml$^{-1}$) to block SLC44A2 were compared to non-immune rabbit IgG (Abcam; 20 µg ml$^{-1}$) to explore the influence of SLC44A2 on neutrophils to bind to either VWF-'primed' platelets or isolated/activated $\alpha_{IIb}\beta_3$.

Isolated PMNs and PBMCs were perfused through channels coated either directly or indirectly with $\alpha_{IIb}\beta_3$, or BSA at 50 s$^{-1}$ for 15 min. Antibodies specific to the different types of leukocytes were added to isolated leukocytes, that is anti-CD16 (RRID:AB_2016663) conjugated to allophycocyanin (APC) to identify neutrophils, anti-CD14-APC (RRID:AB_314190) for monocytes, anti-CD3-APC (RRID:AB_314066) for T-cells and anti-CD19-APC (RRID:AB_314242) for B-cells.

To visualize NETosis, neutrophils were labeled with 8 µM Hoechst dye (cell permeable) and 1 µM Sytox Green (cell impermeable) and monitored for 2 hr. As indicated, isolated PMNs were preincubated with 20 µM TMB-8 (Ca$^{2+}$ antagonist and protein kinase C inhibitor; Sigma), for 15 min, or 30 µM DPI (NADPH oxidase inhibitor; Sigma) for 30 min at 37°C prior to NETosis assays. In some experiments, neutrophils were captured on microchannels coated with anti-CD16 and stimulated with 160 nM PMA prior to analysis of NETosis in the presence and absence of inhibitors.

To confirm the presence of NETs, neutrophils that were captured by activated $\alpha_{IIb}\beta_3$ and fixed with 4% paraformaldehyde after 2 hr. Fixed neutrophils were permeabilized with 0.1% Triton X-100 in PBS for 10 min, blocked with 3% BSA in PBS and, thereafter, incubated with rabbit polyclonal anti-citrullinated H3 (RRID:AB_304752, 10 µg/ml) overnight at 4°C (*Martinod et al., 2013*). Neutrophils were incubated with a goat anti-rabbit secondary antibody conjugated with Alexa647 (Abcam, 1:500) and with the Hoechst dye (8 µM) for 2 hr, washed and then visualized by confocal microscopy.

Quantitation of platelet rolling, aggregation and intracellular $Ca^{2+}$ release was achieved using SlideBook 6.0 software (RRID:SCR_014300). The number of leukocytes rolling/attaching per minute at $50\ s^{-1}$ was derived by counting the number of cells in one field of view over a period of 13 min. NETosis was quantified by determining the proportion of all neutrophils in the microchannel that had undergone NETosis after 2 hr.

## Transcriptomic profiling of human leukocytes

RNA sequencing data from different leukocytes were obtained from the BLUEPRINT consortium (Grassi et al., 2019). For this, neutrophils and monocytes were isolated from peripheral blood. PBMCs were separated by gradient centrifugation (Percoll 1.078 g/ml) whilst neutrophils were isolated by CD16 positive selection (Miltenyi) from the pellet, after red blood cell lysis. PBMCs were further separated to obtain a monocyte-rich layer using a second gradient (Percoll 1.066 g/ml) and monocytes further purified by CD14-positive selection (Miltenyi) after CD16 depletion. For neutrophils and monocytes, gene expression was tested also on Illumina HT12v4 arrays (accession E-MTAB-1573 at arrayexpress). The purification of naive B lymphocytes, naive CD4 lymphocytes, naive CD8 lymphocytes used in this study has been extensively described. Regulatory CD4 lymphocytes (T regs) were isolated by flow activated cytometry using the following surface markers combinations: CD3+ CD4+ CD25+ CD127low. Cell type purity was assessed by flow cytometry and morphological analysis. RNA was extracted using TRIzol according to manufacturer's instructions, quantified using a Qubit RNA HS kit (Thermofisher) and quality controlled using a Bioanalyzer RNA pico kit (Agilent). For all cell types, libraries were prepared using a TruSeq Stranded Total RNA Kit with Ribo-Zero Gold (Illumina) using 200 ng of RNA. Trim Galore (v0.3.7) (http://www.bioinformatics.babraham.ac.uk/projects/trim_galore/) with parameters '-q 15 s 3 –length 30 -e 0.05' was used to trim PCR and sequencing adapters. Trimmed reads were aligned to the Ensembl v75 human transcriptome with Bowtie 1.0.1 using the parameters '-a –best –strata -S -m 100 -X 500 –chunkmbs 256 –nofw –fr'. MMSEQ (v1.0.10) was used with default parameters to quantify and normalize gene expression. Differential gene expression analyses were performed: mature neutrophils (n = 7) vs monocytes (n = 5) and CD4-positive/αβ T cells (n = 8) vs monocytes (n = 5). Regulatory T cells ($T_{reg}$, n = 1) and native B cells (n = 1), are included in the heatmap, for comparison but were not used in differential gene expression analysis due to the low number of biological replicates. We selected genes that were expressed significantly higher in neutrophils than in monocytes, and also those that were significantly higher in CD4-positive/αβ T cells than in monocytes. Their intersection identified 750 genes (598 of which protein coding). From these 598 genes, we selected the 93 genes that contained the Uniprot annotation of 'INTRAMEMBRANE DOMAIN' or 'TRANSMEM DOMAIN'. The effective log2(FPKM+1) data were presented in the heatmap. Further selection involved discarding those transmembrane proteins that are not present on the extracellular membrane, or primarily associated with intracellular membranes as determined by Uniprot annotation. Proteins that (where known) had extracellular regions of <30 amino acids, as determined in Uniprot, that might be less likely capable of mediating specific ligand binding were also excluded. Finally, analysis of proteomic data from the ImmProt (http://immprot.org) resource was used to verify higher levels of protein of each selected gene in neutrophils than in monocytes.

## Expression of SLC44A2 in HEK293T cells

The mammalian expression vector, pCMV6-Entry containing the human *SLC44A2* cDNA C-terminally fused to tGFP was purchased from OriGene. To introduce the rs2288904 SNP encoding a R154Q substitution, site-directed mutagenesis was performed using the R154Q 'top' and 'bot' primers (see Key Resources Table). Successful introduction of the SNP was confirmed by sequencing.

HEK293T cells (RRID:CVCL_0063) were cultured as adherent layers, in humidified incubators at 37°C, 5% $CO_2$, in minimum essential media (Sigma) supplemented with 10% FBS, 1 U/ml Penicillin 0.1 mg/ml Streptomycin, 1% non-essential amino acids (Sigma) and 2 mM L-glutamine. Conditioned media from HEK293T cells were tested using the Venor GeM mycoplasma detection kit (Sigma) and confirmed to be mycoplasma free. Cells were authenticated by out-sourced short tandem repeat analysis (NorthGene) of genomic DNA extracted from HEK293T cells using PureLink Genomic DNA kit (Invitrogen) and confirmed to be HEK293T cells.

HEK293T cells were seeded in 6-well plates 24 hr prior to transfection and transfected using Lipofectamine 2000 (Invitrogen). Transfection efficiency was estimated visually by fluorescent microscopy and quantified using flow cytometry. In all cases, transfection efficiency was >75%. Cells were harvested 24 hr post-transfection with Tryplex (Life Tech) to obtain a single-cell suspension. Cells were washed with complete medium and cells resuspended in serum-free OptiMEM (Life Tech) until use. Transfected HEK293T cell lysates were harvested for Western blot analysis using an anti-tGFP monoclonal antibody (RRID:AB_2622256) at 0.3 µg ml$^{-1}$ to verify expression of the fusion proteins, and the consistent and uniform fusion of tGFP to SLC44A2 variants.

## Flow assays using HEK293T cells

Microchannels were coated with FL-VWF or $\alpha_{IIb}\beta_3$ (coupled via the anti-$\beta_3$/LIBS2 antibody). Thereafter, unlabeled plasma-free blood was perfused over the FL-VWF coated channels at high shear for 3.5 min to capture a layer of 'primed' platelets. Platelet coverage was monitored in bright-field. Channels were subsequently washed with 1xHT buffer to remove the blood and SLC44A2-tGFP or SLC44A2(R154Q)-tGFP transfected HEK293T cells were perfused at low shear (25 s$^{-1}$) for 10 min. Transfected HEK293T cells were also perfused through FL-VWF coated channels (in the absence of platelets for 30 min at 25 s$^{-1}$) to examine any direct interaction with VWF. Transfected HEK293T cells were also perfused through $\alpha_{IIb}\beta_3$ (coupled via the anti-$\beta_3$/LIBS2 antibody) channels at 25 s$^{-1}$ for 10 min. Binding of fluorescent HEK293T cells was quantified by counting the number of cells attached after 10 min across the whole channel and then expressing this as the mean number of cells/field of view. In separate experiments, the ability of GR144053 to block $\alpha_{IIb}\beta_3$, or antibodies that recognize the second or first extracellular loop of SLC44A2, respectively, rabbit anti-SLC44A2 #1 (RRID:AB_2827953) or rabbit anti-SLC44A2 #2 (RRID:AB_2827954) (0–20 µg ml$^{-1}$) to block SLC44A2 were compared to non-immune rabbit IgG (Abcam; 20 µg ml$^{-1}$).

## Genotyping

To identify individuals homozygous for the *SLC44A2* rs2288904-A SNP or for the common/wild-type allele rs2288904-G, 25 µl blood was taken by pin prick from healthy volunteers that provided written informed consent that was approved by the Imperial College Research Ethics Committee (approval reference 19IC5523). Genomic DNA was extracted using PureLink Genomic DNA kit (Invitrogen). DNA yield was quantified by NanoDrop. Genomic DNA from each volunteer was used as a template to PCR amplify a 410 base pair fragment of the *SLC44A2* gene spanning the SNP site using SLC44A2 'top' and 'bot' primers (see Key Resources Table). After amplification, samples were separated by agarose gel electrophoresis and the 410 bands excised, purified using the Gel Extraction kit (Qiagen) and sequenced using the 'top' PCR primer. PMN isolated from genotyped individuals were subsequently used to examine their ability to bind both activated $\alpha_{IIb}\beta_3$ (captured using the LIBS2, anti-$\beta_3$ antibody) and VWF-'primed' platelets, as described above.

## Statistics

Statistical analysis was performed using Prism 6.0 software (RRID:SCR_002798). Differences between data/samples was analyzed using unpaired two-tailed Student's t-test or Mann-Whitney, as appropriate and as indicated in figure legends. Data are presented as mean ± standard deviation, or median ±95% confidence interval. The number of individual experiments performed (n) is given in each legend. Values of p<0.05 were considered statistically significant.

## Acknowledgements

This work was funded through by grants from the British Heart Foundation (FS/15/65/32036 and PG/17/22/32868) awarded to JTBC, KJW and IIS-C. MF is supported by the British Heart Foundation (FS/18/53/33863). The authors declare no competing financial interests. The authors thank Ying Jin and My Dang (Imperial College London) for technical assistance and blood sampling. The authors thank Prof Heyu Ni and Dr Miguel Neves (University of Toronto) for helpful discussions into identification of $\alpha_{IIb}\beta_3$ binding partners.

## Additional information

### Funding

| Funder | Grant reference number | Author |
|---|---|---|
| British Heart Foundation | FS/15/65/32036 | Isabelle I Salles-Crawley<br>Kevin Woollard<br>James TB Crawley |
| British Heart Foundation | PG/17/22/32868 | Isabelle I Salles-Crawley<br>Kevin Woollard<br>James TB Crawley |
| British Heart Foundation | FS/18/53/33863 | Mattia Frontini |

The funders had no role in study design, data collection and interpretation, or the decision to submit the work for publication.

### Author contributions

Adela Constantinescu-Bercu, Data curation, Formal analysis, Validation, Investigation, Visualization, Methodology; Luigi Grassi, Mattia Frontini, Resources, Data curation, Formal analysis; Isabelle I Salles-Crawley, Conceptualization, Data curation, Formal analysis, Supervision, Funding acquisition, Investigation, Visualization, Methodology, Project administration; Kevin Woollard, James TB Crawley, Conceptualization, Formal analysis, Supervision, Funding acquisition, Methodology, Project administration

### Author ORCIDs

Adela Constantinescu-Bercu (iD) http://orcid.org/0000-0002-1274-2867
Kevin Woollard (iD) https://orcid.org/0000-0002-9839-5463
James TB Crawley (iD) https://orcid.org/0000-0002-6723-7841

### Ethics

Human subjects: Specific ethical approval was obtained from the Imperial College Research Ethics Committee (19IC5523) for drawing blood from healthy volunteers and genotyping these for the SLC44A2 SNP. Fresh blood was collected from consented healthy volunteers.

### Decision letter and Author response

Decision letter https://doi.org/10.7554/eLife.53353.sa1
Author response https://doi.org/10.7554/eLife.53353.sa2

## Additional files

### Supplementary files

- Transparent reporting form

### Data availability

All data generated or analysed during this study are included in the manuscript and supporting files. The source data underlying Figs 1, 2, 3, 4, 5c, 6, 8, and Figure 1, 3 and 7 Supplements are provided in separate 'Source Data' files.

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
