## [Decision Letter]

**Acceptance summary:**

This work shows that polymorphonuclear cells (PMNs) and T cells can bind to activated integrin α_IIb_β_3_ and that, under shear, the interaction leads to PMN netosis. The authors identify SLC44A2 as the leukocyte ligand for α_IIb_β_3_ and show that a Q154R polymorphism, a risk factor for thrombosis, greatly favors the interaction of SLC44A2 with the integrin.

**Decision letter after peer review:**

Thank you for submitting your article "Activated α_IIb_β_3_ on platelets mediates flow-dependent NETosis via SLC44A2" for consideration by *eLife*. Your article has been reviewed by three peer reviewers, including David Ginsburg as the Reviewing Editor and Reviewer #1, and the evaluation has been overseen by Satyajit Rath as the Senior Editor. The following individuals involved in review of your submission have agreed to reveal their identity: Mark H Ginsberg (Reviewer #2); Paul Kubes (Reviewer #3).

The reviewers have discussed the reviews with one another and the Reviewing Editor has drafted this decision to help you prepare a revised submission.

Summary:

This work shows that PMNs and T cells can bind to activated integrin α_IIb_β_3_ and that, under shear, the interaction leads to PMN netosis. The authors identify SLC44A2 as the leukocyte ligand for α_IIb_β_3_ and show that a Q154R polymorphism, a risk factor for thrombosis, greatly favors the interaction of SLC44A2 with the integrin.

Essential revisions:

1) The human neutrophil experiments employ too few patients to convincingly make the case that the polymorphism is responsible for the increased adhesion. Knock in mice would help (and could also be used to document differential sensitivity to VTE). Alternatively, consider using a neutrophil model such as differentiated HL60 cells in which SLC44A2 was deleted and rescued with either 154R or 154Q SLC44A2.

2) How is SLC44A2 regulated? Is it increased on plasma membranes or are there affinity changes? Similar surface expression of 154R or 154Q SLC44A2 in transfected HEK293 cells should be demonstrated. Note: GFP expression would not be sufficient unless authors also show that all GFP is associated with SLC44A2 e.g. by Western blot.

3) LPS has been shown to induce platelet-neutrophil interactions without P-selectin release or aggregation of platelets (Clark et al., 2007). Does LPS affect SLC44A2 binding to α_IIb_β_3_?

4) The authors claim there is no role for releasate from platelets to get NET production, as NETosis can be induced with purified α_IIb_β_3_. However, this took more than an hour and not all neutrophils produced NETs. Is it possible that releasate expedites and increases NET production? It’s hard to dismiss releasate based on the α_IIb_β_3_ data.

---

## [Author Response]

Essential revisions:1) The human neutrophil experiments employ too few patients to convincingly make the case that the polymorphism is responsible for the increased adhesion. Knock in mice would help (and could also be used to document differential sensitivity to VTE). Alternatively, consider using a neutrophil model such as differentiated HL60 cells in which SLC44A2 was deleted and rescued with either 154R or 154Q SLC44A2.

We accept that the experiments from healthy genotyped donors shown in Figure 8F involves comparatively few individuals and that, by themselves, do not make the case for the contribution of the SLC44A2 polymorphism to reduced binding to platelets/activated α_IIb_β_3_. This was limited by the pool of healthy consented donors available to us, as well as the practicality of timing of call backs to fit in with the nature of the experimentation performed on each sample – each sample needs to be prepared fresh for immediate experimentation limiting experiments to 2 donor samples prepared and analysed in parallel per day through channels with accompanying controls. Despite the numbers, however, the data/differences between genotype were very clear and we firmly believe that they complement the accompanying data using HEK293T cells transfected with SLC44A2(R154) or SLC44A2(Q154) shown in Figure 8C-E that further support the involvement of the polymorphism in mediating platelet binding. We have now also added these data with those from the R154/Q154 heterozygous individuals presented in new Figure 8F and added this finding to the text in the Results section which further support our findings.

We absolutely agree with the reviewers that SLC44A2 R154Q knock-in mice would prove a valuable tool to explore this, and this is certainly on our radar. The global SLC44A2 knockout mice have recently been shown to be protected against experimental DVT supporting the role of SLC44A2 – these data are accepted/in press so a full reference will be available presently. The contribution of the polymorphism to DVT remains untested in the murine system (which have an Arg at position 154 of SLC44A2). Of note the, VLRDGDCPAVLI sequence in SLC44A2 containing R154 is very highly conserved between species, consistent with a conserved functional role for this part of the extracellular loop. Naturally, the generation of the knock-in mice will take some time, and this is beyond the scope of the current study or revision, but something that we would like to explore in future experiments.

The deletion/ablation of SLC44A2 expression from HL60 using CRISPR-Cas9 technology followed by the specific selection of cells in which SLC44A2 deletion has been successful is potentially an interesting idea. However, this represents a significant amount of work within the time frame of a revision, for potentially only marginal gain with the potential for further complication (see below), as we have already shown the involvement of SLC44A2 in transfected non-myeloid cell lines (Figure 8C-E). We specifically used HEK293 cells because they do not normally express SLC44A2 on their surface, which provides a clean system to probe the contribution of the SLC44A2 receptor and the two different genotypes to the interaction with α_IIb_β_3_. We were highly encouraged by the clear nature of the data presented in Figure 8C-E, which, in conjunction with the genotyped volunteer experiments, justifies the conclusions that we draw, in our opinion.

The use of CRISPR-Cas9 approaches to knock out SLC44A2 requires appreciable work to both identify and select the cell populations that have SLC44A2 deleted prior to rescue. Simple knock down with siRNA would be inadequate due to the potential lifespan of SLC44A2 on the surface of the cells and the lack of knockdown in daughter cells as the cells divide. An additional potential issue with this approach is maintaining the selected HL60 cell line in a phenotype akin to circulating PMNs. The potential for the cells to become stimulated/activated during processing to the point where they may present other adhesion molecules, or activate cell surface integrins, which may further complicate the data obtained from these approaches.

2) How is SLC44A2 regulated? Is it increased on plasma membranes or are there affinity changes? Similar surface expression of 154R or 154Q SLC44A2 in transfected HEK293 cells should be demonstrated. Note: GFP expression would not be sufficient unless authors also show that all GFP is associated with SLC44A2 e.g. by Western blot.

SLC44A2 gene regulation has not been explored. It is clear that SLC44A2 expression is robust in resting neutrophils as is presented in Figure 7 and Figure 7—figure supplement 1. SLC44A2 is reportedly primarily and preferentially present on the plasma membrane, but has also been detected in certain neutrophil granule membranes (Rørvig S, et al. J Leukoc Biol. 2013;94(4):711-21). It is currently not known whether there are any affinity changes or different conformations that selectively interact with different ligands. Whether SLC44A2 expression levels change appreciably in response to different agonists, and the timeframe over which this occurs has not to our knowledge been reported.

With respect to the surface expression/tagging of SLC44A2 with GFP, the SLC44A2(R154)-GFP and SLC44A2(Q154)-GFP fusion proteins, GFP is tagged to the intracellular C-terminus. We purchased verified SLC44A2 cDNA fused to turbo GFP. Please note that the delay in resubmission of this manuscript was due to our oversight that the GFP fusion was turbo GFP, rather than EGFP – consequently, the anti-EGFP Abs that we purchased did not recognise turbo GFP fusion proteins. We have amended reference to EGFP through the manuscript to replace with tGFP. Only upon recognising this oversight were we able to source an anti-tGFP mAb that we now use. We provide Western blot data to show the comparable detection of tGFP using an anti-tGFP mAb associated with SLC44A2. This also confirms that the tGFP signal is only associated with the SLC44A2 fusion protein in both R154 and Q154 genotypes, and there is no evidence of proteolytic removal of tGFP as revealed by the lack of a band at ~26kDa (MW of turbo GFP). We provide this blot as an inset in Figure 8C and have referenced this blot in the Results section of the manuscript and also into the Material and methods section.

3) LPS has been shown to induce platelet-neutrophil interactions without P-selectin release or aggregation of platelets (Clark et al., 2007). Does LPS affect SLC44A2 binding to αIIbβ3?

This is a very interesting idea, and something that we too hypothesised (i.e. that LPS might exert a similar priming effect upon platelets that enabled them to interact with leukocytes rather than transforming them into procoagulant platelets). There are conflicting data on the influence of LPS upon platelets, with some data suggesting a priming effect, akin to VWF binding under flow such as reported by Clark and colleagues, 2007, and other purporting more robust activation e.g. Lopes Pires et al., 2017, which we reference along with others. Although we have not formally explored this ourselves, a priming-type effect seems more likely. A major issue with such experiments though, is that physiologically, the interaction between LPS on intact bacteria and TLR4 on platelets is, by itself, insufficient to withstand the forces of flow present within the vasculature. This means that platelets will not encounter LPS in isolation (i.e. without another agonist being present – such as collagen, thrombin, fibrin and/or VWF). This makes extrapolation of such experiments in which platelets are incubated with LPS alone under static conditions redundant in the in vivo setting.

We have not formally explored or quantitatively assessed the influence of LPS upon the binding of SLC44A2 to α_IIb_β_3_. We did perform a few preliminary experiments in which platelets were captured on flow channels coated with VWF and thereafter neutrophils and LPS were flowed over these surfaces for 10 mins. Channels were washed in buffer further supplemented with LPS and then visualised under static conditions for 2 hours. In these experiments, we saw no difference in neutrophil capture/binding to the platelet covered surface in the presence or absence of LPS. We also did not detect any difference in NETosis over the time frame of analysis. Based on these preliminary findings, we did not explore this further.

Of note, we also performed flow experiments over whole bacteria coated on microchannel surfaces. In these experiments, we observed VWF-dependent platelet recruitment to the bacterial surfaces (that was further augmented if we inhibited ADAMTS13), and also platelet-dependent neutrophil recruitment. Under these conditions, we observed very rapid NETosis. We provide an example of these data Author response image 1.

These data, although related, are part of a separate study and suggest that the exposure of neutrophils to both “primed” platelets and bacteria leads to a more profound and rapid generation of NETs. At this time, we have not formally determined the contribution of LPS to the process, but we have observed similar effects on both Gram-negative and Gram-positive bacteria, suggesting that LPS is not necessarily an essential driver of this synergy. This will be submitted in a follow up study.

**Author response image 1. sa2fig1:** *E. coli* labelled with BacLight (red) were adsorbed to the surface of microchannels (left panels). Thereafter, whole blood supplemented with DiOC_6_ to label platelets and leukocytes was perfused at 1000s^-1^ for 3 mins and then at 50s^-1^ for 3 mins. Platelets bound to *E. coli* in a VWF-dependent manner. Inhibiting ADAMTS13 (which cleaves VWF) with a pAb significantly increased platelet binding was observed, demonstrating the VWF-dependence of the platelet capture. Conversely when both ADAMTS13 and VWF were inhibited almost no platelet binding was detected, again highlighting the dependency upon VWF. When the flow was reduced to 50s^-1^, leukocytes bound to bound platelets, but not when VWF had been blocked (not shown).

4) The authors claim there is no role for releasate from platelets to get NET production, as NETosis can be induced with purified αIIbβ3. However, this took more than an hour and not all neutrophils produced NETs. Is it possible that releasate expedites and increases NET production? It’s hard to dismiss releasate based on the αIIbβ3 data.

We absolutely agree with the reviewers that platelet releasates may potentially further contribute to NETosis, our contention is that the platelet releasate is not necessary for NETosis to occur in neutrophils captured via SLC44A2 binding to α_IIb_β_3_. This is a subtle, but important distinction when considering this process, particularly when thinking about the early events in DVT. Previous studies have alluded to the potential contribution of platelet releasate components to NETosis (Stark et al., Blood 2016, Gollomp et al., JCI Insight 2018). However, the binding of platelets to VWF alone (i.e. during the early stages of DVT) does not by itself cause appreciable degranulation as we show in Figure 2—figure supplement 1 – this is of course different when other agonists become available, but these are invariably not present during the early stages of DVT. Moreover, the effects of flow have the potential to rapidly remove/dilute any released granule contents. This is diminished if a thrombus forms which greatly reduces solute transport out of the thrombus. Therefore, we do not dispute the potential for other agonists, including those derived from platelet granules (such as HMGB1, PF4, RANTES) to promote NETosis, but rather we contend that they are not the sole mechanism by which NETosis can be induced. We would also like to highlight that the timeframe for NETosis by neutrophils bound either to VWF-primed platelets or to purified activated α_IIb_β_3_ is very similar, and we have now highlighted this in the text. In these experiments, platelets were capture/primed on VWF for 3 mins, prior to perfusion of neutrophils for 10 mins. Thereafter neutrophils were analysed for 2 hours. Under these conditions, if any degranulation were to synergistically augment NETosis, we would have expected either an increased number of neutrophils NETosing, or an increased rate at which NETosis was detected on primed platelets. To clarify this point further, we have modified the text in both the Results and the Discussion sections.

In summary, it is our opinion that the VWF priming represents part of the tunable response of platelets. In the setting of targeting of bacteria we propose that the priming is sufficient to promote neutrophil recruitment, but by itself is not sufficient to transform platelets into procoagulant platelets leading to thrombus formation in the absence of overt vessel damage.